# Development of smart anti-glycan reagents using immunized lampreys

Tanya R. McKitrick [1], Christoffer K. Goth [1,3], Charles S. Rosenberg [2], Hirotomo Nakahara[2], Jamie Heimburg-Molinaro[1], Alyssa M. McQuillan[1], Rosalia Falco[1,4], Nicholas J. Rivers [1,5], Brantley R. Herrin[2,6], Max D. Cooper [2] & Richard D. Cummings[1✉]

Studies on the expression of cellular glycans are limited by a lack of sensitive tools that can discriminate specific structural features. Here we describe the development of a robust platform using immunized lampreys (*Petromyzon marinus*), which secrete variable lymphocyte receptors called VLRBs as antibodies, for generating libraries of anti-glycan reagents. We identified a wide variety of glycan-specific VLRBs detectable in lamprey plasma after immunization with whole fixed cells, tissue homogenates, and human milk. The cDNAs from lamprey lymphocytes were cloned into yeast surface display (YSD) libraries for enrichment by multiple methods. We generated VLRB-Ig chimeras, termed smart anti-glycan reagents (SAGRs), whose specificities were defined by microarray analysis and immunohistochemistry. 15 VLRB antibodies were discovered that discriminated between linkages, functional groups and unique presentations of the terminal glycan motif. The development of SAGRs will enhance future studies on glycan expression by providing sequenced, defined antibodies for a variety of research applications.

[1] Department of Surgery, Beth Israel Deaconess Medical Center, National Center for Functional Glycomics, Harvard Medical School, Boston, MA 02215, USA. [2] Department of Pathology and Laboratory Medicine, Emory Vaccine Center, Emory University School of Medicine, Atlanta, GA 30322, USA. [3] Present address: University of Copenhagen Glycomics Program, Copenhagen, Denmark. [4] Present address: Marine Science Center, Northeastern University, Boston, MA 02115, USA. [5] Present address: University of Alabama Birmingham, Birmingham, AL 35294, USA. [6] Present address: Acceleron Pharma, Boston, MA 02110, USA. ✉email: rcummin1@bidmc.harvard.edu

Glycans are present in the glycocalyx of all living cells[1] where they comprise a major component of glycoprotein mass[2]. They are fundamentally important for development of complex organs and multi-cellular organisms, and play crucial roles in diverse biological processes, including cell communication, adhesion, migration and protein structural stability. Atypical glycosylation patterns frequently arise in malignant tissues, and are associated with cancer invasion and progression[3]. However, while it is widely recognized that carbohydrates play essential roles, the underlying spatial and temporal expression profile of glycans is difficult to discern, primarily due to the paucity of reliable reagents.

Carbohydrate expression is explored using lectins, but such reagents have limitations, such as broad specificity[4–6]. While monoclonal antibodies (mAbs) are more robust reagents, their availability toward glycan targets is limited and immunization of rodents with human or rodent glycans can have unpredictable success. Due to similarities in the murine and human glycome, many glycans can be weak antigens and immunization frequently results in the generation of IgM isotypes, which are often cross reactive with other glycans[5,7–9]. In addition, given the number of different glycan determinants predicted to be present in animal glycomes[10], it would be difficult to cover even a fraction of these glycans with conventional mAb technologies. Thus, it is necessary to develop rapid and high throughput methods for generating specific glycan-binding reagents to decipher the expression of the human glycome.

We explored an alternative approach for the generation of glycan-specific reagents which is to leverage the unique immune system that evolved in the jawless vertebrates[11–14]. The sea lamprey (*Petromyzon marinus*) expresses a novel family of highly diverse proteins composed of leucine-rich repeats (LRRs) termed variable lymphocyte receptors (VLRs) as antigen receptors[15,16]. Upon antigenic stimulation, VLRB cells clonally expand and differentiate into plasma cells that secrete circulating antigen-specific VLRB proteins. Such VLRBs are generated by recombinatorial assembly of hundreds of partial LRR gene segments, and this gene conversion-like assembly mechanism is capable of generating >$10^{14}$ distinct receptors, comparable to the human Ab repertoire[17]. Assembled VLRB genes encode a single polypeptide chain that forms a 15–25 kDa crescent-shaped protein, consisting of multiple LRR motifs that form a continuous beta-sheet lining the concave surface that comprises the antigen-binding domain[13,18].

Here we explore the capability of lamprey to generate anti-glycan antibodies, and delineate the methodology for translating this response into reagents that can be broadly used for traditional immunological research applications. While prior studies demonstrated that immunization of lampreys with cells can generate anti-glycan responses to select carbohydrate antigens[11–14,19–26], the breadth of this response has never been explored. Lampreys were immunized with a diverse set of cellular immunogens and the resulting plasma was screened on a microarray containing ~600 unique glycan structures. For each immunization, the resulting VLRB repertoire was expressed as a stable YSD library where we pursued different enrichment strategies, including glycan microarrays, labeled whole cells and lysates, to isolate distinct populations of anti-glycan antibodies. After enrichment, we defined a panel of mAbs prepared as Ig chimeras termed smart anti-glycan reagents (SAGRs), which have unique specificity and provide unprecedented possibilities to accurately dissect the expression and localization of distinct glycan epitopes. In particular, the combinatorial application of these SAGRs will advance current analytical capabilities, and highlights the potential of these novel reagents. The stable genetic library can be screened by glycan microarray technologies and other outlined approaches to rapidly identify, enrich, and ultimately produce recombinant SAGRs that recognize specific glycan determinants.

## Results

### Anti-glycan VLRBs specific to wild-type and mutant CHO cells.

For immunization we chose four CHO cell lines that have distinct glycomic profiles to immunize lamprey using paraformaldehyde-fixed (~$10^6$) cells, which were directly injected into the intracoelomic cavity without adjuvant at 2-week intervals (3×). Two weeks post-final injection, plasma from immunized and naive animals was collected and screened on the Consortium for Functional Glycomics (CFG) glycan microarray, which contains over 600 unique glycan structures. Bound VLRBs were detected with anti-VLRB murine IgG (4C4)[27] and Alexa-Fluor anti-mouse IgG secondary. As a guiding hypothesis, we considered that some glycan antigens might be present in the material that are below the typical detection by glycomic analyses, yet sufficient to induce antibody responses. In any case, it is important to note that no complete glycome of any mammalian cell has been defined, and only abundant glycan structures are often characterized.

Plasma from each CHO cell-immunized lamprey demonstrated relatively specific anti-glycan responses (Fig. 1). Wild-type CHO Pro-5 cells generated VLRBs that recognize poly-LacNAc (-3Galβ1-4GlcNAcβ1-)$_n$ chains (Fig. 1a), consistent with reported glycan profiling that Pro-5 express complex N-glycans containing poly-LacNAc repeats (www.functionalglycomics.org). Immunizations with Lec8 CHO cell mutants generated VLRBs predominantly to glycans with Galβ1-3GalNAc (core 1), and lacked a response to poly-LacNAc chains (Fig. 1b). This is interesting, because while Lec8 cells are deficient in galactose due to a 97% deficiency of the UDP-galactose transporter[28], they can express low levels of specific types of galactose-containing glycans, including some glycolipids and glycosaminoglycans[29].

In addition, we immunized with two engineered Lec8 cell lines that had been transfected with the *Caenorhabditis elegans* β1,4-N-acetylgalactosaminyltransferase (Lec8GT) to generate the Lacdi-NAc (LDN) antigen (GalNAcβ1-4GlcNAc-R) and Lec8GTFT cells that co-express the *C. elegans* β1,4-N-acetylgalactosaminyltransferase and human α1,3-fucosyltransferase IX to generate fucosylated LDN structure LDNF (GalNAcβ1-4(Fucα1,3)GlcNAc-R)[30]. Lec8GT cells generated VLRBs against glycans expressing terminal GalNAc residues, including LDN and weakly to LDNF (Fig. 1c). Immunization with Lec8GTFT cells generated a stronger VLRB response to LDNF (Fig. 1d), and resembled the glycan-binding profile observed in Lec8GT cells.

All immunizations with wild-type and CHO cell mutants generated VLRBs against fucosylated (i.e., GalNAcβ1-3(Fucα1-2)Gal-R and Fucα1-2Galβ1-4(6s)GlcNAc-R) and 3-O-sulfated galactose. While it is unknown if CHO Pro-5 and Lec8 mutants express either FUT1-2 or ST3GALT1-4, these glycosyltransferases have been measured in other CHO cell mutants[31]. Unsurprisingly, when comparing the overall binding profile of different CHO cell mutant immunization on CFG arrays, the Lec8GT and Lec8GTFT elicited the most similar anti-glycan response (Pearson $r = 0.6768$, Supplementary Fig. 1a) while wild-type CHO Pro.5 and Lec8GTFT had the least similar repertoire (Pearson $r = 0.3186$, Supplementary Fig. 1a). Overall, these results demonstrate that immunization with CHO cell constructs can generate specific anti-glycan VLRBs against the unique glycan signature of each cell line.

### Glycan-specific VLRBs to pig lung, Tn4 cells and human milk.

To further explore anti-glycan VLRB responses against tissues, we

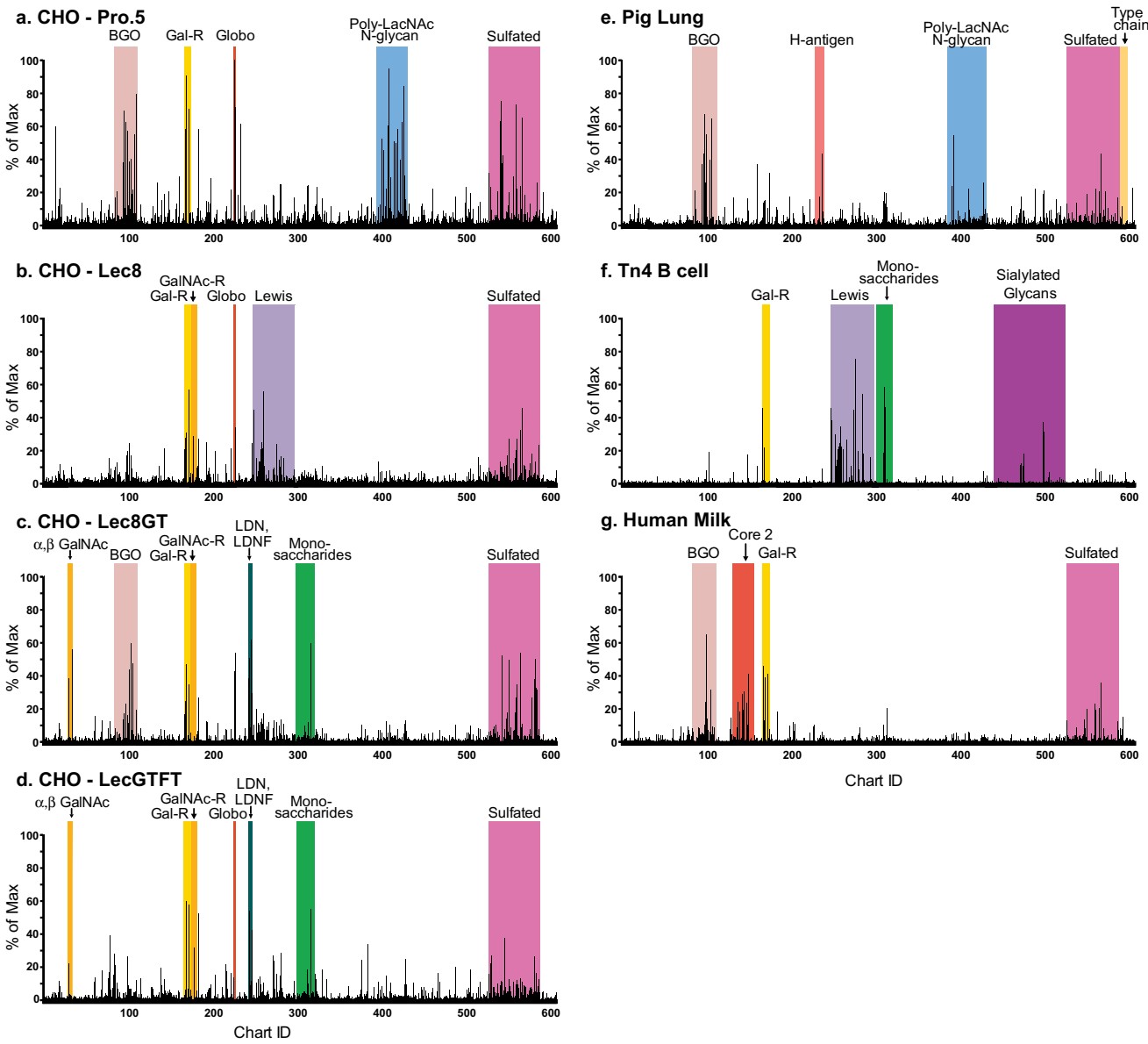

**Fig. 1 VLRB binding profiles of immune stimulated plasma screened on the CFG glycan microarray reveal that lampreys mount a targeted and diverse anti-glycan response against the given antigen.** Plasma was assayed at a 1:2 dilution for all samples except for the Tn4 B cells (f), which was screened at 1:5. Detection was achieved with an anti-VLRB mouse antibody, and goat anti-mouse Fc-488. The immunogens injected into the lamprey were **a** CHO—Pro.5, **b** CHO—Lec8, **c** CHO—Lec8GT, **d** CHO—Lec8GTFT, **e** Pig lung, **f** Tn4 B cells, and **g** human milk.

homogenized a portion of a pig lung into a single cell suspension and directly immunized the lampreys. This immunization generated anti-glycan VLRBs against predicted epitopes in swine tissue, including the Galα1-3Gal[32] epitope, α2-3 and α2-6 linked sialylated N-glycans[33] and α2-6 sialylated core 1 and core 2 O-glycans[34]. The predominant VLRB response against pig lung was against the type 2 H-antigen (Fucα1-2Galβ1-4GlcNAcβ1-R) on both N- and O-glycan presentations, presumably due to expression of the pig AO blood group system[35]. Interestingly, there was a large diversity of VLRBs that recognized sulfated glycans and unusual sequences such as Galβ1-3GlcNAcβ1-3Galβ1-4GlcNAcβ1-6(Galβ1-3GlcNAcβ1-3)Galβ1-4Glc, a branched glycolipid (Fig. 1e).

Lampreys immunized with a human B cell line (Tn4) derived from a person whose leukocytes express the Tn antigen[36], produced a diverse repertoire of VLRBs recognizing the Lewis blood group antigens (Fig. 1f). Specifically, to Lewis X (Galβ1-4

(Fucα1-3)GlcNAc-R) and Lewis A (Galβ1-3(Fucα1-4)GlcNAc-R), the T antigen (Galβ1-3GalNAc-R), and sialylated core 2 structures (Neu5Acα2-6Galβ1-4GlcNAcβ1-3Galβ1-4GlcNAcβ1-6(Galβ1-3) GalNAcα-R). There has been no investigation characterizing the N- and O-glycan profile of this cell line, so it is difficult to predict the expected immune response. The glycome of healthy human B cells has been characterized by MALDI-TOF, and results suggest healthy B cells express a wide range of oligomannose, bi-antennary, tri-antennary and complex-type N-glycans and potentially Lewis antigens (www.functionalglycomics.org). However, since this cell line was derived from an individual whose leukocytes express the Tn antigen due to hypermethylated promoter of the *Cosmc* gene[36], the cell surface glycome would presumably be quite different from the glycome of a healthy B cell. We tested this serum on our Tn glycopeptide array[37], and observed a unique binding pattern to the Tn antigen with poly-GalNAc-Ser residues (Supplementary Fig. 2).

 COMMUNICATIONS BIOLOGY | https://doi.org/10.1038/s42003-020-0819-2

The results of many immunizations suggest that cellular presentation is important to initiate immune responses in lampreys, since injecting purified soluble proteins has had mixed results[22,24]. Thus, it was unclear if we could initiate a robust lamprey immune response with soluble sugars. We approached this question by immunizing with unmodified human milk, which is a complex mixture of lipids, soluble proteins, many unique free glycans and maternal epithelial cells, leukocytes and beneficial bacteria[38]. Lamprey were immunized five times with 20 μl of whole milk and the resulting plasma, contained VLRBs bound to previously described free human milk oligosaccharides (Fig. 1g). This includes VLRBs recognizing the type 3 H-antigen (Fucα1-2Galβ1-3Gal-NAcα-R), type 1 H-antigen (Fucα1-2Galβ1-3GlcNAcα-R), 2′-fucosyllactose (Fucα1-2Galβ1-4Gal-R), Galβ1-2Gal-R, Galβ1-3Gal-R, Neu5Acα2-6Galβ1-4GlcNAcβ1-6(Galβ1-3)GalNAc-R, and other glycans. Naive plasma lacked significant anti-glycan responses (Supplementary data 1), confirming that immunization results in production of immunogen-specific antibodies.

All of the lamprey immunization profiles on the CFG array are presented in a heat map (Supplementary Fig. 1b) and the complete data set can be found in Supplementary data 1. Overall, each biological sample stimulated a relatively unique anti-glycan VLRB profile, but we observed some common binding patterns. Most samples contained detectable VLRBs to Galβ1-3GalNAcβ1-R, a structural disaccharide that is the essential backbone of all ganglio series of glycolipids expressed by most vertebrates. In addition, VLRBs against the type 2 H-antigen, 2′-fucosyllactose, and sulfated Gal or GlcNAc residues were common, implying that such antigens are present in many biological samples and are highly immunogenic to lamprey[19].

### Lamprey vs. mouse Ab response to cellular and viral antigens.
Historically, mice have been the prototypical model system to generate mAbs to protein and glycan antigens. However, investigators have had difficulties in generating a broad repertoire of anti-glycan mAbs due to the limitations inherit within the murine immune system, such as self-tolerance[17,39]. We set out to more directly compare murine vs. lamprey systems for generating anti-glycan antibodies, by immunizing with human type AB erythrocytes and inactivated Simian immunodeficiency virus (SIV) particles by three separate intraperitoneal injections every 2 weeks (3×). Two weeks after the final immunization, serum (mice) and plasma (lamprey) were collected and screened on the CFG glycan microarray to detect carbohydrate-specific mouse IgGs, IgMs, and VLRBs (Fig. 2). We also analyzed mouse serum collected prior to immunization to examine the preexisting repertoire of IgG and IgM antibodies.

Immunized mice generated only minor anti-glycan IgG responses to the antigens within this time frame (Supplementary data 1), but we did observe detectable IgM responses (Fig. 2a–d). Both mice contained circulating anti-glycan IgM antibodies in their serum prior to immunization, as has been observed previously[40,41]. The repertoire of these IgM antibodies for both animals analyzed were highly correlated ($r = 0.88$), and were against a variety of blood group and Lewis antigens to name a few (Fig. 2a, b). After immunization with type AB erythrocytes and SIV particles, the murine anti-glycan IgM response included antibodies that bound to all three type 2 blood groups (BGH: Fucα1-2 Galβ1-4GlcNAc-R, BGA:GalNAcα1-3(Fucα1-2)Galβ1-4GlcNAc-R, and BGB: Galα1-3(Fucα1-2)Galβ1-4GlcNAc-R), as well as Le$^x$ antigen (Fig. 2c, d). Murine serum from SIV immunized animals also had IgMs recognizing poly-LacNAc N-glycans and some sulfated glycans (Fig. 2d). However, many of these IgMs were detected within serum pre-immunization, so it is difficult to conclude which antibodies were specific for the

immunogen. The plasma from comparably immunized lamprey was quite distinct from the murine IgM profile (Fig. 2e, f). Overall, the lamprey plasma from both immunizations contained VLRBs against the BGH, BGA and BGB and Lewis antigens, and a variety of sulfated and sialylated glycans. While many of the same classes of glycans were recognized by both VLRBs and mIgMs (i.e., BGB antigens), the lamprey VLRBs bound to distinct presentations and isomeric configurations of these motifs. For example, the top five BGB glycans bound by mIgM present in murine serum all contained the type 2 backbone (Galα1-3(Fucα1-2)Galβ1-4GlcNAc-R). Whereas the top five BGB glycans bound by VLRBs present in lamprey plasma were of the type 1 moiety (Galα1-3(Fucα1-2)Galβ1-3GlcNAc-R). The relationship between the antibody responses in mice vs. lamprey is plotted in the form of a force graph[42] (Fig. 2g), where each individual node (colored, A–F) represents the different samples analyzed. Each gray circle represents an individual glycan, and all the RFU values below 1000 were omitted. Uniquely bound glycans will protrude off of the node and will not be connected to any other sample. This figure demonstrates that lampreys generated several VLRBs that recognize unique glycans that were not bound by mouse serum and vice versa. For both data sets, the VLRB and IgM profiles were highly divergent for type AB erythrocytes (Fig. 2h, $r = 0.2386$) and SIV particles (Fig. 2h, $r = 0.1485$). The complete array data set can be found in Supplementary data 1.

The results suggest that the lamprey is at minimum a complementary model system to a mouse for generating anti-glycan-binding reagents. While both the lamprey and mouse produced anti-glycan antibodies after immunization, the advantage of the lamprey system lies within the downstream processing of the reagents. While the creation of hybridomas has been historically used to generate mAbs in mice, it can be expensive and time consuming to screen against a large repertoire of glycans. Alternatively, the murine immunoglobulin cDNA library could be expressed using Fab phage display or as an scfv YSD library; however, both approaches have technical limitations to consider. Phage display may bias the library toward antibodies that can be folded properly in bacteria and are not amenable to enrichment by flow cytometry or glycan microarrays. Murine scfv YSD libraries can be constructed from both naive and/or immunized B cells; however, due to the dimeric nature of immunoglobulin proteins, one might lose the specificity of the antibodies generated during the immunization series due to the random pairing of heavy and light chain. Both phage and YSD approaches require a complete amplification of the heavy and light variable segments which requires a multiplex of over 50 primers for successful amplification[43,44]. However, because lamprey VLRBs are a single gene product, amplification of the VLRB repertoire can be completed using a single set of primer pairs, as well as providing permanent cDNA libraries encoding all lamprey antibodies induced from the immunization.

### Pairing microarrays with YSD to enrich for anti-glycan VLRs.
In order to identify, enrich for, and ultimately characterize glycan-specific VLRB proteins from immunized lamprey, we used a YSD expression system to generate libraries of antibodies (Fig. 3). The fundamentals of growing yeast cultures, creating YSD libraries and the method of induction are described in detail in previous publications[44,45]. Briefly, the YSD library was constructed by co-electroporating the linearized expression vector (pCT-ESO-BDNF)[22] and VLRB cDNA library into the EBY100 strain of Saccharomyces cerevisae[46]. The end result is a YSD library whereby the VLRB is tethered to the cell surface via fusion with the endogenous Aga1p and Aga2p cell surface proteins and a myc tag localized at the C-terminus (Fig. 3b).

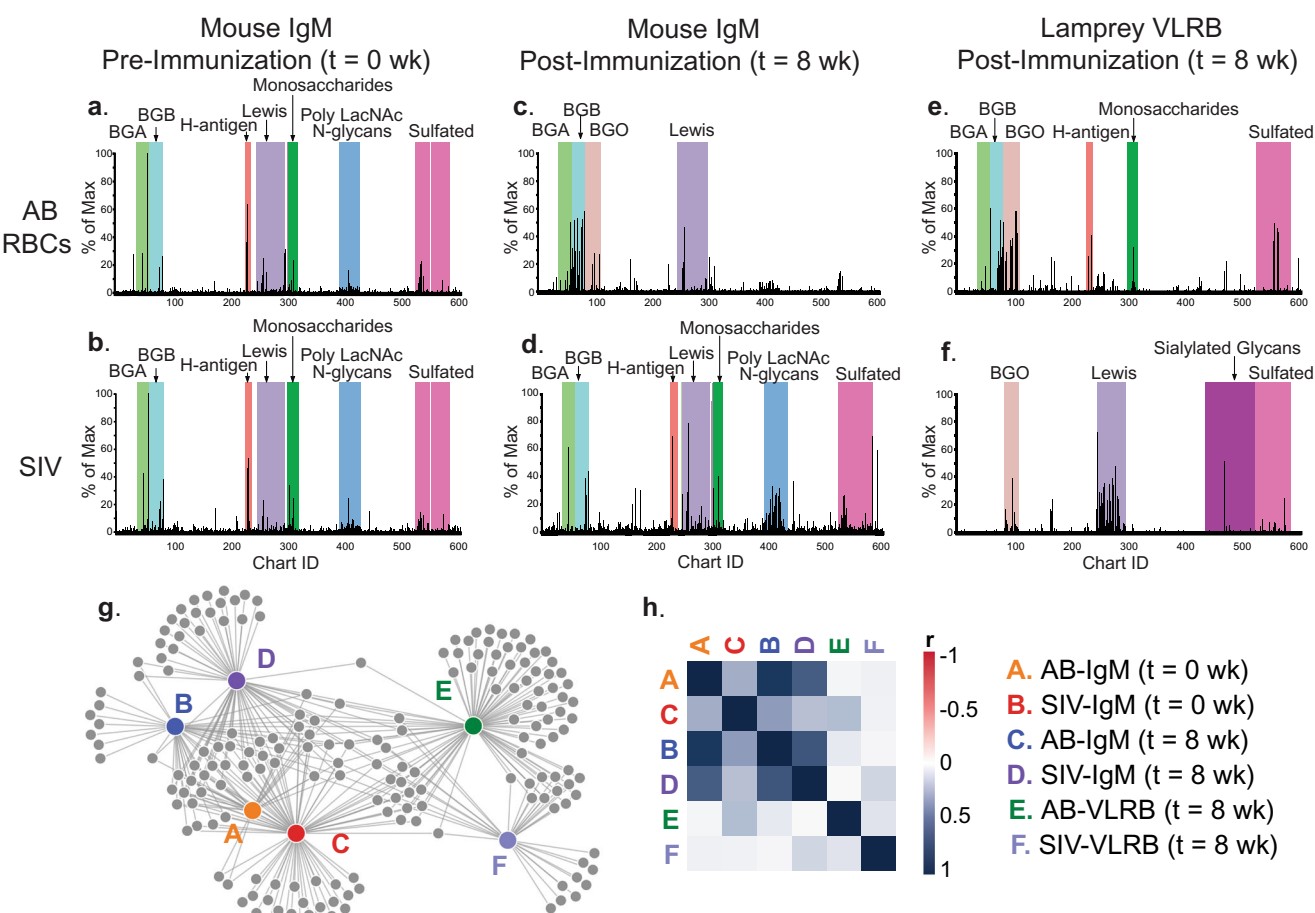

**Fig. 2 Lamprey elicit a distinct VLRB response from that of a mouse model when challenged with the same immunogen.** Mouse serum prior to the immunization ($t = 0$ wk) was collected and analyzed on the CFG array (**a**, **b**). Lamprey and mice were immunized with type AB human erythrocytes and inactivated SIV particles three times at 2-week intervals. Serum was isolated from both lamprey and mice 2 weeks after the final immunization ($t = 8$ wk), and screened on the CFG glycan microarray. The binding profile on the CFG are shown for mouse IgM (**c**, **d** 1:50 dilution), and lamprey VLRB (**e**, **f** 1:10 dilution). The network of common and unique binders is represented by a force graph (**g**) and the Pearson $r$ correlation analysis between lamprey VLRB and mouse IgM binding profiles for each sample (**h**).

Initially, we aimed to use the glycan microarrays to enrich the YSD libraries for glycan-specific targets, and needed to establish that the yeast would bind to the glycan microarrays via the VLRBs and not the cell surface endogenous yeast proteins[47]. To test this, we used VLRB (RBC36) which is known to be specific for the type 2 H-antigen[13,25]. We expressed RBC36 both as a VLRB-mIgG fusion protein (Fig. 3a, left), and on the surface of yeast to determine the specificity on the CFG array. The binding profile of the RBC36-mFc chimeric protein is depicted in Fig. 3a (right), and as predicted, is specific to glycans that contain the type 2 H-antigen motif. We incubated the clone expressing RBC36 on the array and after extensive washing, we observed macroscopically that the yeast were bound to the array in patterns of six replicates (Fig. 3c), which is the printing configuration for individual glycans on the CFG array. We then labeled the bound yeast with an anti-myc-488 mAb and determined that the RBC36-yeast clone retained similar specificity to the soluble protein (Fig. 3b).

We then developed the methodology to isolate the glycan-specific clones found within the unsorted Tn4 B cell library, using the CFG array as the template. After an overnight incubation at 4 °C, the non-enriched library contained only a few yeast clones which were visible macroscopically, but were undetectable by the microarray scanner. We transferred the bound yeast colonies to

solid media via replica plating, and then to liquid growth media. This process was repeated three times and each stage was monitored on the CFG array. This technique resulted in successive enrichment of the library for VLRBs that recognize glycans with the type 2 H-antigen, sialylated and sulfated determinants (Fig. 3d). The complete array data for enrichment and yeast bound to the array can be found in Supplementary data 2.

**Type O erythrocytes elicit a diverse repertoire of VLRBs**. As to be expected, the plasma obtained from lampreys immunized with type O libraries contained VLRBs primarily against the type 2 BGO and sulfated H-antigen glycans, and at first glance the profile appeared to be fairly uniform (Fig. 4a). To assess the true diversity of anti-glycan targets within this library, we used two different strategies to enrich for endogenous cell surface glycan targets on type O human red blood cells (RBC) and then incubated the libraries on the CFG array. First, we labeled the cell surface proteins of with biotin, lysed the labeled RBCs and incubated the lysate with the non-enriched library (Fig. 4b). The antigen-specific clones bound to biotinylated erythrocyte glycoproteins were then enriched by MACS using streptavidin-coated magnetic beads and subsequently enriched by FACS using the

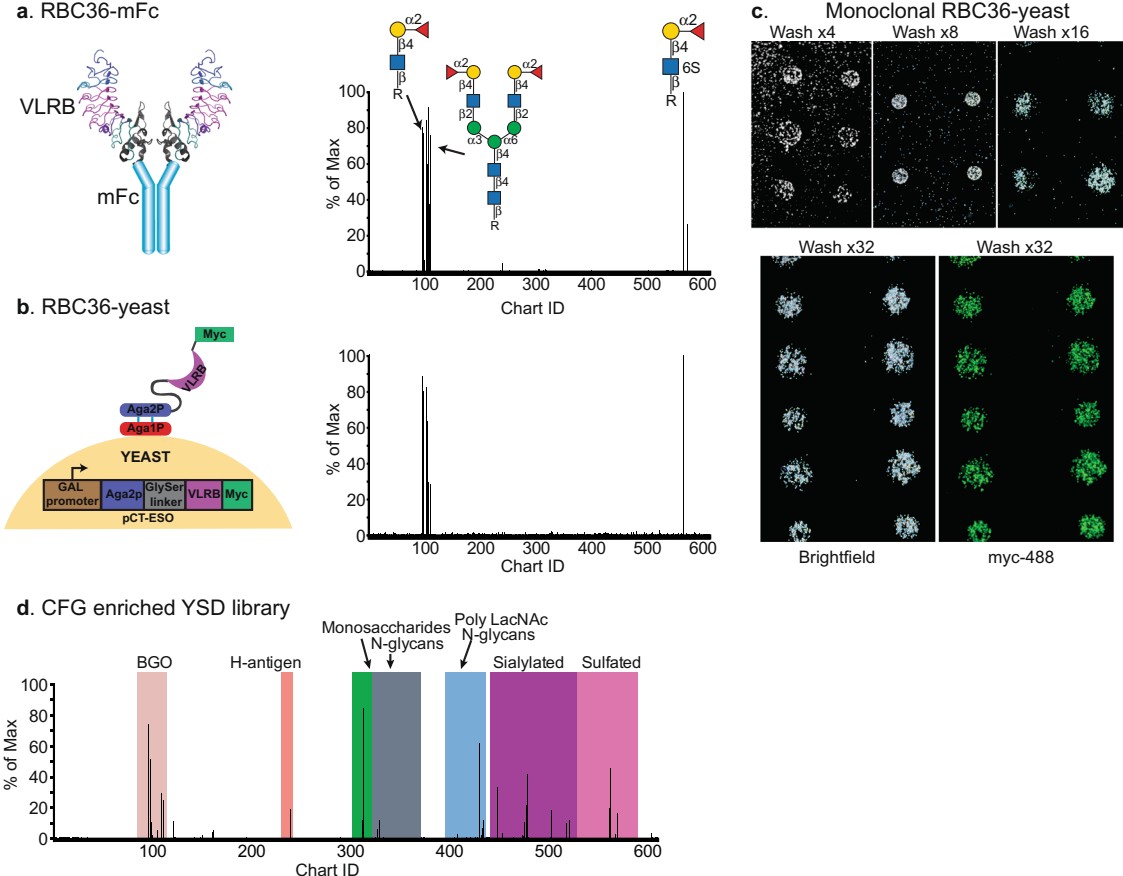

**Fig. 3 Pairing yeast surface display and glycan microarray technology has led to the rapid identification of VLRB binding specificity and development of a novel enrichment strategy of yeast surface display libraries. a** Monoclonal VLRB RBC36 was expressed as a mouse Fc chimeric protein and screened on the CFG array. **b** Monoclonal VLRB RBC36 expressed on the surface of yeast retains the same binding specificity as the soluble protein. **c** Macroscopic images of yeast bound to the glycan microarray during the wash steps (top panel, left to right) and stained with an anti-Myc-488 antibody (bottom panel). **d** Three rounds of enrichment using the CFG array identified VLRBs which bound to distinct glycan epitopes.

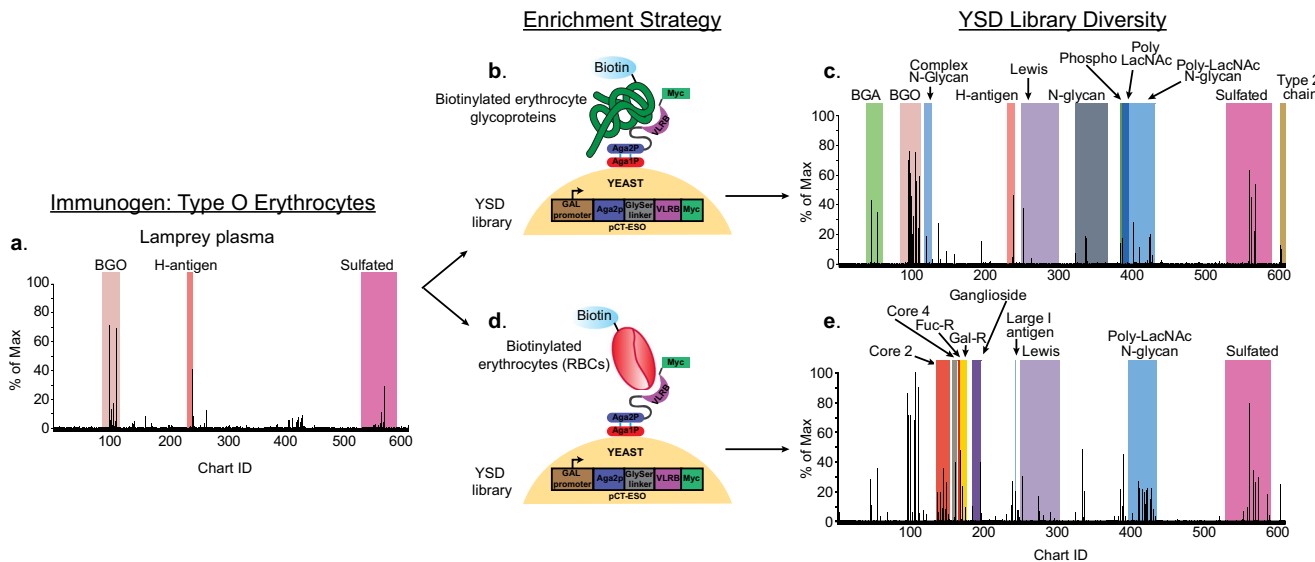

**Fig. 4 Immunization with type O erythrocytes revealed a predominate VLRB response against the type 2 H-antigen. a** Type O plasma tested on the CFG array. The YSD library was incubated with biotinylated cell lysate (**b**), enriched by MACS and FACS and the diversity of this enrichment method is determined by incubation on the CFG array (**c**). Alternatively, the YSD library was labeled with biotin-labeled whole cells (**d**) enriched by MACS and FACS, and incubated on the CFG array (**e**).

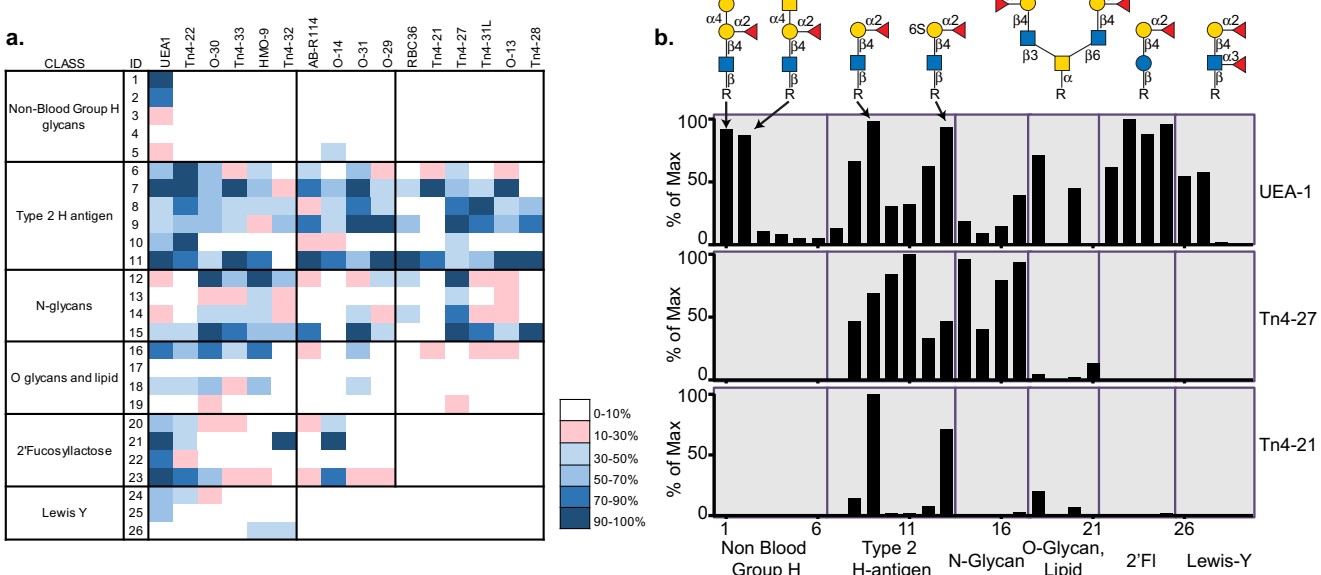

**Fig. 5 We have identified and characterized 12 monoclonal VLRBs from four YSD libraries which bind to glycans carrying the type 2 H-trisaccharide motif, and compared their specificity with previously reported type 2 H VLRBs and the plant lectin UEA-I.** The RFU values were normalized to a scale of 0–100 and are shown in the form of a heat map (**a**). The RFUs are plotted for two VLRBs, Tn4-27 that preferentially binds to the type H antigen on N-glycan presentations, and Tn4-21 that binds to O-glycan presentations of the antigen (**b**).

same method. Predictably, the YSD library that was enriched with lysate contained many VLR clones bound to type 2 H-antigen glycans, 2′ fucosyllactose and sulfated versions of these structures. In addition, we detected VLRBs bound to the Le[y] antigen, complex N-glycans Galβ1-4GlcNAcβ1-2Manα1-6(Galβ1-4GlcNAcβ1-2Manα1-3)Manβ1-4GlcNAcβ1-4GlcNAc and Galβ1-4GlcNAcβ1-2Manα1-6(Galβ1-4GlcNAcβ1-4(Galβ1-4GlcNAcβ1-2)Manα1-3) Manβ1-4GlcNAcβ1-4GlcNAc, as well as poly-LacNAc motifs on both linear and N-glycan presentations (Fig. 4c).

As an alternate strategy, we enriched the YSD library using biotinylated intact RBCs instead of lysate. (Fig. 4d). The yeast complexed with the cells were then enriched by MACS and FACS as previously described. By contrast, the library enriched with intact cells contained VLRBs that bound to many of the same glycans found within the library enriched with lysate (Pearson $r$ = 0.7118), but possessed a broader diversity of positive binders, including clones that recognized glycolipids and O-glycans (Fig. 4e). In particular, we found VLRBs that bound to the large I blood group (Galβ1-4GlcNAcβ1-6(Galβ1-4GlcNAcβ1-3)Galβ1-4GlcNAc), core 2 (Galβ1-4GlcNAcβ1-6GalNAc) and core 4 structures (Galβ1-4GlcNAcβ1-3Galβ1-4GlcNAcβ1-6(Galβ1-4GlcNAcβ1-3Galβ1-4GlcNAcβ1-3)GalNAc).

**Expression and characterization of type 2 blood group H SAGRs.** Using a combination of the enrichment strategies outlined above, we identified twelve new VLRB sequences isolated from four different immune libraries (Type O RBC, Type AB RBC, Tn4 B cells and HMO). Each of the VLRBs were chosen to explore based on sequence similarity to three other VLRBs, RBC36, O13, and Tn4-22, whose specificity has been previously reported[13,25], and thus we predicted that these VLRBs would bind to similar glycan structures. Sequence alignment of all fifteen of these proteins revealed that the majority of the amino acid polymorphisms are located in the N-terminus (LRRNT-LRRV4) and are nearly identical from LRRV5 to the LRRCT (Supplementary Fig. 3). All 15 VLRBs were expressed as SAGRs, screened on the CFG glycan microarray and the results are depicted in Fig. 5a. The glycans presented in this heat map only include

positive binders for the H-antigen VLRBs and the lectin *Ulex europaeus* agglutinin (UEA-I) as a positive control. For the complete binding profile on the CFG array see Supplementary data 3, and the glycan structures included in the heat map in Fig. 5a are drawn in Supplementary Fig. 4.

As we predicted, all fifteen of the VLRBs bound to structures containing the type 2 H-antigen and did not cross react with either the type 1 (Fucα1-2Galβ1-3GlcNAc) or type 3 (Fucα1-2Galβ1-3GalNAc) motifs. However, several VLRBs bound to 2′-fucosyllactose and Le[Y] and had a more similar binding profile to the UEA-I plant lectin. Of the 15 VLRBs characterized, 6 were able to discriminate between the type 2 H-antigen and 2′-fucosyllactose, glycans which differ only by a single acetamide group. Even within the six VLRBs specific for the type 2 H-antigen, some were able to discriminate between distinct presentations of the terminal motifs. For example, Tn4-21 has a limited binding profile, and recognizes only six structures on the CFG array (Fig. 5b), which are linear and O-glycan presentations. Alternatively, Tn4-27 binds to nearly all of the linear and N-glycan presentations of the antigen, but does not bind the glycan when presented as an O-glycan.

In summary, each of these VLRBs has a unique amino acid sequence and specificity. Even though all bind to the same motif, no two VLRBs bind to the exact same presentations of the antigen. The type of linker that couples the sugar to the slide, and whether the H antigen is presented as a linear structure or on branched N- and O-glycans influences the binding specificity of each VLRB. Interestingly, while UEA-I is the most commonly used reagent in the field to study blood group H, the specificity of the lectin is quite broad with high intensity staining for a few non-blood group H antigens, 2′fucosyllactose and Le[Y] (Fig. 5b). This broad specificity limits the utility of this reagent when probing biological samples, as the investigator cannot reliably conclude which glycan is bound by the lectin without orthogonal validation. The VLRBs reported here uniquely bind to subsets of the glycans recognized by UEA-I and are able to discriminate subtle differences in glycan presentation, which highlights the versatility and power of this system to develop highly specific anti-glycan reagents.

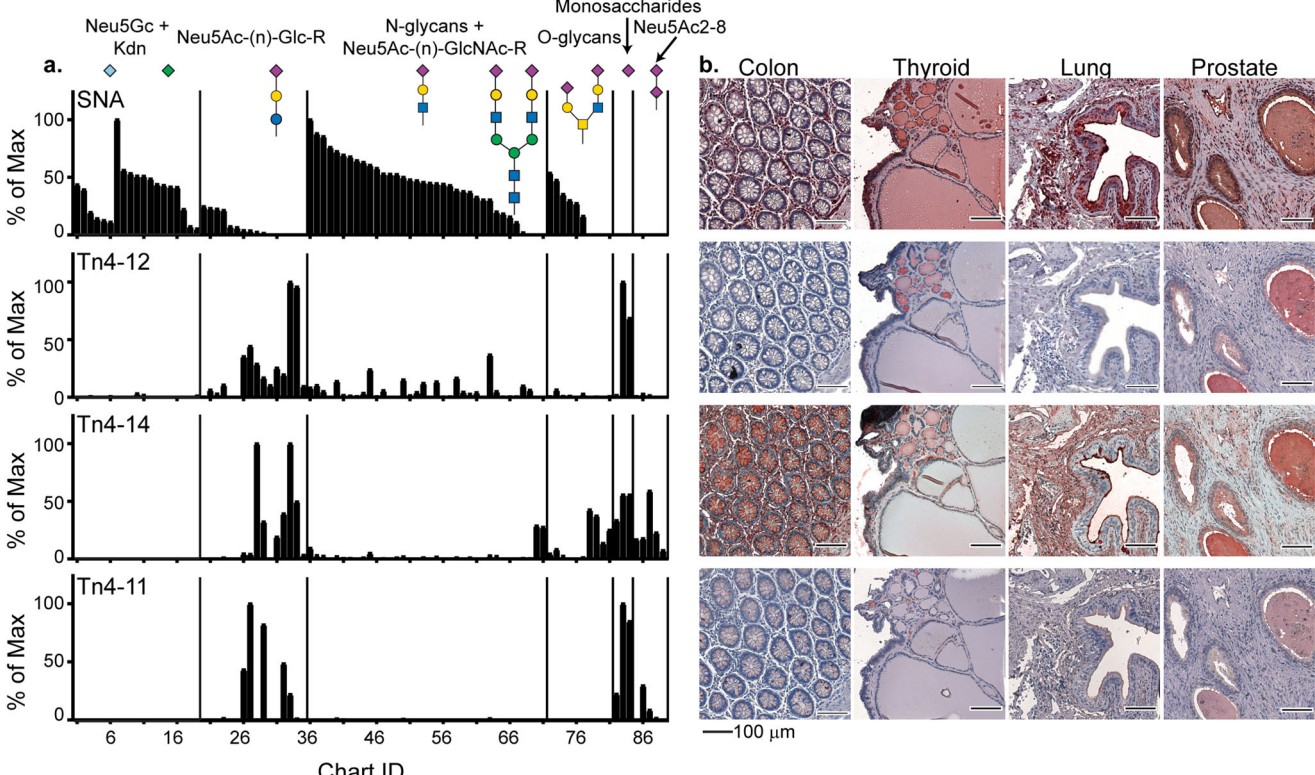

**Fig. 6 VLRBs specific for sialic acid demonstrate a distinct binding profile from the plant lectin *Sambucus nigra* (SNA). a** A comparison of the binding profiles of 3 monoclonal VLRBs on the CFG and NCFG sialyl derivative array to SNA. All of the glycans bound by SNA and the VLRBs terminate with a 2–6-linked sialic acid and the RFU values were normalized to a scale of 0–100. **b** Commercial human tissue arrays were probed with VLRBs and clear differences were observed in staining patterns for colon, thyroid, lung, and prostate demonstrating that these antibodies can be used to differentiate between subsets of sialylated glycans in immunohistochemistry applications. All images of tissue slides contain 100 μm scale bars located in the bottom right corner.

**Identification of VLRBs specific for sialylated glycans**. From the CFG enriched Tn4 B cell library, we have identified three VLRBs which bind to glycans with either terminal α2-6 or α2-8 sialic acid linkage. Similar to the H-antigen-specific VLRBs, each of these amino acid sequences are identical from the LRRV3 to the LRRCT and the majority of the observed polymorphisms are found in the LRRNT, LRR1, and LRRV1. Despite the high degree of sequence similarity between the VLRs (80–90%), the glycan microarray binding profiles of these three proteins are only moderately correlated (Supplementary Fig. 5). Without a glycan: VLRB crystal structure, it is not possible to determine exactly which amino acid polymorphism is responsible for the variation in glycan recognition, however, we can make structural predictions as to which amino acids are facing the antigen binding pocket. A comparison of the sequence alignment suggests that only five amino acid polymorphisms located in LRRV1 are driving the observed differences in specificity.

To determine the specificity of Tn4-11, -12, and -14, the recombinant VLRB-Fc fusion proteins were screened on the CFG array and the NCFG Sialyl Derivative array[48] at 5 μg/ml (Fig. 6a). The binding profiles were compared with the commonly used plant lectin *Sambucus nigra* (SNA), which is known to bind broadly to terminal 2–6 sialic acid (and 2–3 with weaker affinity). Strikingly Tn4-11, -12, and -14 showed almost no overlap with the SNA binding (Fig. 6a), but instead bound to other 2–6 sialylated structures. More specifically, Tn4-14 showed binding to the cancer-associated sialyl Tn structure and to the 2–8 linkage present in polysialic acid. Tn4-11 and Tn4-12 showed more similar binding patterns, although with different preference for sialylated type-1 and -2 chains, respectively (Fig. 6a). Tn4-12 showed slightly broader binding patterns in general, while Tn4-11

showed some reactivity toward 2–8 linked sialic acids (see Supplementary data 3 for complete data). In addition, we probed normal human tissue macroarrays and observed clear differences in staining patterns for colon, thyroid, lung, and prostate (Fig. 6b). The specificity of the VLRBs bound to the tissue arrays were further clarified by the elimination of binding after neuraminidase treatment (Supplementary Fig. 6). Taken together, the combination of these sialic acid binding SAGRs clearly provides a mechanism to accurately dissect the precise structure of sialylated glycans in different tissues.

## Discussion

Here we have described our development of a robust screening and enrichment approach to identify and characterize lamprey-derived VLRB mAbs specific for glycan antigens (Fig. 7). The success here builds upon earlier studies demonstrating that some VLRBs recognize carbohydrate antigens and can be expressed using a YSD platform[20,22]. The generation of anti-glycan VLRB-YSD libraries, and the screening and enrichment approach presented here is complementary, but in many ways superior, to conventional approaches using murine hybridoma screening or screening by single chain antibodies in phage display. Like classical Ig-based mAbs, VLRBs bind to antigens with high affinity and specificity[12,24,25,49], and they have several practical advantages over Ig-based mAbs. VLRB proteins are composed of a single peptide chain, rather than Ig heavy and light chains, and therefore individual sequences can be easily PCR amplified from a cDNA library. The VLRB antigen-binding domain is modular and relatively small (15–25 kDa vs. 50 kDa Ig Fab′), and therefore can be easily fused to other proteins while retaining its specificity.

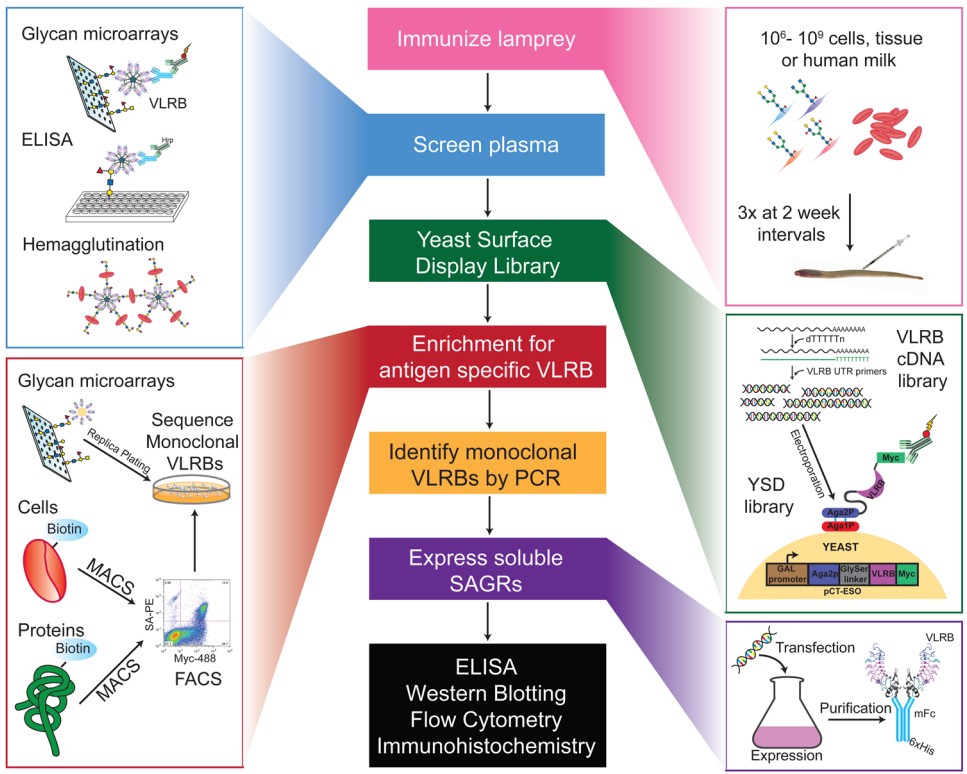

**Fig. 7** Flow chart depicting the entire method of the generation of SAGRs.

The crescent-shaped, β-sheet-based antigen-binding surface, along with the extended LRRCT loop, may allow VLRB proteins to bind to epitopes that are inaccessible to the relatively flat antigen-binding surface of most Ig-based Abs[50].

A key advantage is the stable and cloned nature of the YSD libraries, which can be used repeatedly for screening and enrichment of anti-glycan specificities. Using the YSD library, we have developed several methods to isolate very minor yeast clones within the expressing anti-glycan antibodies. The multiple strategies for enrichment is only possible with this approach, as these are stable libraries and unlike hybridomas, there is no time limit for identifying desirable clones. In fact, the YSD libraries are a renewable source of antibodies to glycan antigens and limited only by the glycans available for screening and characterization of specificity. The readily available sequence encoding each VLRB also allows easy engineering of the sequences for producing novel mutants with altered specificities and novel chimeric proteins.

Lampreys last shared a common ancestor with mammals ~550 million years ago, and this evolutionary distance may provide the potential for the generation of VLRB proteins that recognize highly conserved proteins and carbohydrates that are not immunogenic in rodents[18]. These reagents have proven useful in a number of critical biological assay platforms, such as glycan microarray, flow cytometry, western blotting, and in probing tissue in immunohistochemistry assays[51,52]. VLRB proteins also have a wide variety of applications in biomedical research, including inhibition of cell signaling pathways[53,54], as scaffolds for drug delivery[55], and as diagnostic reagents[11,21,23]. Such reagents in their Ig chimeric form, which we have termed Smart Anti-Glycan Reagents or SAGRs, should be invaluable in the field of glycan studies, as they will facilitate spatial and temporal expression studies of glycans in biological systems.

These discoveries may help address a key deficiency in the glycosciences, which is the need for robust libraries of reagents that are monoclonal in nature and can recognize the vast diversity of glycan structures expressed in the glycomes of organisms. Historically, while there have been few useful murine monoclonal antibodies to glycan antigens, including the SSEAs[56], cancer biomarkers, e.g., CA19-9[57] and ganglioside GD2[58], there are few murine monoclonals that recognize the predicted thousands of different glycan determinants expressed in the human glycome[10]. Going forward, SAGRs will be invaluable tools for studying glycan expression both temporally and spatially, and when linked to gene expression profiles, they could provide insights into the molecular regulation of cellular glycomes.

It is interesting that sera from lampreys immunized with different immunogens, e.g., cells and homogenates, reveal that each elicits a relatively unique diversity of anti-glycan antibodies. This might be expected as it is commonly assumed that glycans expressed on different glycoproteins, cells and tissues, differ in structure and immunogenic determinants, for which thousands have been predicted[59]. However, while the lamprey-derived SAGRs will be invaluable to the field, there are some limitations, several of which may be overcome in future developments. For instance, the glycome of the lamprey, and the mechanism of self-tolerance in the lamprey, are not defined and could limit the ability of lampreys to generate anti-glycan VLRBs to some antigens. In addition, the ability to isolate anti-glycan VLRBs is limited by the availability of different glycan structures. While the glycan microarrays are a robust screening tool to determine the VLRB specificity, it is a platform that could lead to the identification of false positives due to differences in presentation of the glycan on the array and in a natural tissue. For example, we discovered several VLRBs that appear to discriminate between N- and O-glycan presentations of the type II H antigen. However, when using this reagent in a primary cell or tissue, this interpretation should be independently determined. Like many reagents used in the glycosciences, these experiments should be done in concert with enzymatic treatments and mass spectrometry when possible. In addition, the glycan arrays used here,

e.g., the CFG glycan array of ~610 glycans, are synthetic materials that represent only a fraction of the glycan determinants known and predicted. One possible partial solution to this limitation is the development of 'shotgun glycan microarrays'[60], derived from natural material in cellular glycomes. Thus, such natural glycan arrays could be prepared from cells or tissues as a discovery tool for both novel antibodies and glycans not yet characterized.

In summary the implications of our study and development of SAGRs are substantial. Libraries of anti-glycan antibodies generated through the VLRB technology could help bridge the gap and become complementary tools for mass spectrometry and the identification of glycan structural isomers. SAGRs could also be used for pull-down glycoproteomics and other sensitive methods to identify glycan expression specific proteins and peptides. Finally, there is a need to identify in a spatial and temporal fashion the expression of glycans, which will likely require immunohistochemical approaches, for which SAGRs are ideally suited as demonstrated here. The availability of SAGRs could be transformative toward the end goal of the cell- and tissue-specific expression of glycans in the Human Glycome and to help generate a 'Human Glycome Atlas' of expressed carbohydrate antigens in human tissues.

## Methods

**Antigens.** Peripheral blood samples were collected in heparin coated collection tubes with informed and signed consent from healthy donors at the Vaccine Center at Emory University, Atlanta, GA, under an approved Institutional Review Board protocol. Samples were completely de-identified for this study. Total leukocytes, erythrocytes and plasma were isolated using lymphocyte separation medium (Corning). Erythrocytes were stored in Alsever's solution (NaCl 4.2 g/L, sodium citrate—$2H_2O$ 8 g/L, citric acid—$H_2O$ 0.5 g/L, D-glucose 20.5 g/L) at 4 °C for a maximum of 1 month. Human milk was purchased from the Mothers Milk Bank (Austin, TX). Pig lung was purchased from Lampire Biological Products (Pipersville, PA), homogenized and stored in frozen aliquots prior to injection. Wild-type CHO (Pro.5) and mutant (Lec8) cells were purchased from ATCC and the Lec8GT and Lec8GTFT cell lines were generated as part of a previous publication[30]. Tn4 B cells were described and generated from a previous publication[36,61]. All cell lines were fixed in 4% PFA, washed, and resuspended in 0.66% PBS prior to immunization.

**Lamprey housing, immunization, plasma, and leukocyte isolation.** *Petromyzon marinus* larvae (8–15 cm in length, ~2–4 years in age) were collected from tributaries to Lake Michigan (Lamprey Services, Ludington, MI) and housed in sand-lined aquarium tanks at 20 °C in the AAALAC-accredited animal care facility at Emory University, under an ethics approved Institutional Animal Care and Use Committee protocol. Animals ($n = 3$) were anesthetized with Tricaine-S (0.1 g/L) and given intracoelomic injections ($n = 3$) at 2-week intervals with antigen. Human milk injections were given once per week, for a total of five injections. For immunization, $10^9$ erythrocytes, 20 μL of whole milk, 15 μg of SIV particles as measured by a standard BCA assay, $10^6$ pig lung and $10^6$ of the PFA fixed cell lines were given. Two weeks after the final injection, the animals were sacrificed with Tricaine-S (1 g/L) and exsanguinated. Total lamprey blood was collected into 0.67x PBS with 30 mM EDTA and layered onto 55% Percoll and centrifuged for 20 min at $500 \times g$. Total leukocytes were collected, washed three times in 0.67x PBS and stored in RNAlater at −20 °C. Lamprey plasma was collected and stored at 4 °C and examined for positive VLRB titers. We tested for positive titers by a standard hemagglutination assay (erythrocytes) with lamprey plasma, by ELISA (human milk, SIV particles, and pig lung lysate), flow cytometry (CHO cells and mutants) and by glycan microarrays (Tn4, described below).

**Murine immunizations and serum isolation.** Six- to eight-week-old BALB/cAnNHsd mice were obtained from Harlan (Indianapolis, IN) and maintained at Emory University under standard conditions. Mice were given intraperitoneal injections without adjuvant, at 2-week intervals with $10^9$ type O and type AB human erythrocytes and 15 μg of inactivated SIV particles. In total, three immunizations were given and blood was collected using a submandibular blood collection method. Blood was collected 1 day prior to immunization, and 2 weeks after the final injection. Total blood was allowed to clot, overnight at 4 °C and centrifuged at $400 \times g$ for 15 min. Serum was collected and stored at −20 °C until defrosted for microarray analysis.

**Analysis of lamprey serum and VLRB specificity on the CFG array.** Lamprey plasma was diluted (1:2, 1:5 or 1:10 depending on the sample as indicated in the figures) into TSM buffer (20 mM Tris, 150 mM NaCl, 2 mM $CaCl_2$, 2 mM $MgCl_2$) with 1% BSA and 0.05% tween-20 and incubated on the CFG glycan microarray

containing over 600 unique glycan structures (version 5.0, 5.1 and 5.2 depending on the sample as indicated in the figures). An anti-VLRB mouse IgG mAb called 4C4[15] was used to detect anti-glycan-specific VLRBs present within the lamprey plasma. Monoclonal VLRB recombinant proteins were bound to the CFG glycan arrays at five or ten-fold concentration intervals. Alexa-Fluor 488 labeled goat anti-mouse IgG mAb was used for detection of both lamprey plasma and the VLRB recombinant fusion proteins. Murine serum was diluted 1:50 and assayed for IgG and IgM binding. Alexa-Fluor 488 labeled goat anti-mouse IgG and Alexa-Fluor 647 goat anti-mouse IgM mAb was used for detection in simultaneous assays. A more detailed description of the CFG binding protocol can be found at www.functionalglycomics.org.

**Yeast culture and VLRB-YSD library production.** Briefly, total RNA was extracted from lamprey leukocytes using the RNeasy Plant Mini Kit (Qiagen) and a cDNA library was generated using Superscript III First-Strand Synthesis system (ThermoFisher Scientific) with oligo dTTT primers. VLRB specific cDNA libraries from immune stimulated animals were PCR amplified from 2 μL of total cDNA with VLRB specific primers (VLRB.UTR.F: 5′-CACGCTCTCCGCTACTCG-3′, VLRB.UTR.R: 5′-CATCCCCGACCTTTGCAC-3′). These samples were PCR purified, and amplified a second time with primers that contain 50 bp of sequence homology to the pCT-ESO expression vector (pCETSO.VLRB.F: 5′-GGTGGAGGAGGCTCTGGTGGAGGCGGTAGCGGAGGCGGAGGGTCGGCTAGCGCATGTCCCTCGCAGTG-3′, pCETSO.VLRB.R: 5′-GATCTCGAGCTATTACAAGTCCT′CTTCAGAAATAAGCTTTTGTTCGGGATCCCGTGGTCGTAGCAACGTAG-3′). These amplifications were gel purified. Both amplifications were done with KOD High Fidelity Hot Start polymerase and used the following protocol: 95 °C 2 min, [95 °C 30 s, 55 °C 30 s, 72 °C 30 s] × 40, 95 °C 2 min, 4 °C. Two micrograms of total pCETSO.VLRB PCR products were electroporated into the EBY100 strain of *Saccharomyces cerevisiae* with 1 μg of the linearized expression vector (pCETSO-BDNF). After transformation, the YSD library is transferred into SD-CAA selection media where the size of the library can be approximated (solid media) and non-transformed yeast strains are eliminated (liquid media). To induce VLRB expression on the cell surface, yeast surface display libraries were inoculated into SD-CAA media and grown overnight at 30 °C, shaking at 260 rpm. Libraries were always grown in liquid media containing 10-fold the library size ($1 \times 10^6$ yeast/ml). After ~16 h of growth, yeast cultures were diluted 1:5 in fresh media and grown for an additional 3 h at 30 °C, shaking at 260 rpm. At the end of this incubation, yeast libraries were resuspended in SG-CAA induction media and grown for 18–24 h at 20 °C, shaking at 260 rpm. To determine if the induction was successful, the yeast library was labeled with anti-Myc-488 mAb and monitored by flow cytometry.

**Analysis of VLRB expressing yeast on the CFG array.** After induction, ~$1 \times 10^5$ monoclonal yeast were washed three times in PBS and resuspended in PBS with 0.05% tween and 1% BSA. Yeast were labeled with anti-Myc 488 and incubated on the CFG glycan microarray for a minimum of 1 h at room temperature. Unbound yeast were removed with sequential washes using PBS with 0.05% tween and 1% BSA, PBS and finally water. The slides with the bound yeast were dried, scanned and analyzed according to the standard CFG protocol. Induced YSD libraries were incubated overnight, gently shaking at 4 °C prior to washing and analyzing.

**Tn4 B cell specific VLRB-YSD enrichment strategy.** After induction, ~$1 \times 10^6$ of the non-enriched YSD library were incubated overnight on the CFG microarray, gently shaking at 4 °C in PBS with 0.05% tween and 1% BSA. Slides were washed using the method described above. Bound yeast were transferred to SD-CAA agar plates and grown overnight at 30 °C. Colonies were then transferred to liquid media and induced as described above. This process was repeated three times until individual clones were sequenced and tested for specificity on the array.

**Erythrocyte specific VLRB-YSD library enrichment strategies.** YSD libraries were enriched for Type O erythrocyte specific VLRBs using two different methods. The cell surface proteins on red blood cells were biotinylated according to the manufacturer's instructions using EZ-Link^TM Sulfo-NHS-LC-Biotin (Thermo-Fisher – Pierce). The first method of enrichment involved co-incubating induced YSD libraries with the intact biotinylated erythrocytes for 1 h at room temperature on rotation. The yeast bound to the labeled erythrocytes were then captured on streptavidin labeled magnetic beads (MACS sorting) and eluted directly into SD-CAA media and grown overnight. This library was further enriched by FACS using the same approach. The second method was to co-incubate the induced yeast with the biotin-labeled lysed red blood cells. The success of the MACS and FACS enrichment was monitored by flow cytometry using the anti-Myc 488 and SA-PE secondary reagents. The enriched libraries were transferred onto SD-CAA agar plates in order to sequence and characterize individual monoclonal VLRB clones.

**Generation of VLRB-mIgG chimeric proteins.** Yeast monoclonals of interest were first identified by PCR, using primers which amplify the region upstream from the VLRB sequence (pCTESO.seq.F 5′-CGACGTTCCAGACTACG-3′, pCTESO.seq.R 5′-TACAGTGGGAACAAAGTCG-3′). PCR amplification was done directly from the yeast, using the following protocol: 37 °C for 30 min, 95 °C 2 min, [95 °C 30 s, 58 °C 30 s, 72 °C 30 s] × 35, 72 °C 2 min. The polymerase used for amplification was KOD High Fidelity Hot Start, with the addition of zymolase (1 unit per 60 μL reaction) to

digest the cell wall of the yeast. PCR products were purified and reamplified with VLRB.F.Nhe: 5′-ggctagcCATCATCACCATCACCATGCATG TCCCTCG-3′, VLRB.R.Bam: 5′-gcaggatccCGTGGTCGTAGCAACGT-3′ to add the Nhe1 and BamH1 cut sites onto the product for molecular cloning. PCR products were purified and digested with High Fidelity Nhe1 and BamH1 at 37 °C for 2 h.

The lentiviral expression vector used was purchased from systems bio (pCDH-CMV-MCS-T2A-copGFP CD524-1) and modified to contain the murine IgG2a Fc sequence and the interleukin-2 signal sequence for secretion. Ultimately, after molecular cloning, the expression vector contained the following series: IL2-6xHIS-VLRB-mIgG2a-T2A-GFP. Lentivirus was generated in 293T cells by triple transfection of the pCDH VLRB containing plasmid and the second generation packaging plasmids (psPAX2 and pMD2.G). Transfection was carried out by incubating 30 μg of plasmid DNA (10 μg of each plasmid) with 90 μg of linear polyethylenimine (PEI) for 30 min in 1 ml of serum free media at room temperature. The DNA-PEI complex was added dropwise to 293T cells grown in 10 cm dishes with 10 ml of media and ~70% confluency. After 72–96 h of incubation, the media was harvested, centrifuged at $1000 \times g$ for 4 min and filtered through a 0.45 μm filter. In total, $1 \times 10^5$ of 293F freestyle cells were then incubated for 72 h in the media containing the viral stocks. After 48 h, GFP expression was confirmed microscopically. VLRB-mFc chimeric proteins were purified from the 293F media using a His-Pur Ni-NTA column.

The sequences for each of the monoclonal VLRB proteins characterized within this paper have been deposited into GenBank with the following accession numbers: Tn4-21: MN904726, Tn4-11: MN904727, Tn4-12: MN904728, Tn4-14: MN904729, O-14: MN904730, O-29: MN904731, Tn4-27: MN904732, Tn4-32: MN904733, HMO-9: MN904734, AB-R114: MN904735, Tn4-28: MN904736, Tn4-31L: MN904737, Tn4-33: MN904738, O-31: MN904739, and O-30: MN904740.

**Immunohistochemistry**. Tissue sections were baked at 60 °C for 45 min followed by deparaffinization with xylene and rehydration with graded alcohols. Antigen retrieval was done by heating slides in a pressure cooker for 3 min in citrated buffer (pH 6.0, 10 mM trisodium citrate). After cooling down at room temperature, tissue sections were incubated with 3% hydrogen peroxide for 10 min. For neuraminidase treatment, slides were incubated at 37 °C for 3 h in 5 mM CaCl and 50 mM ammonium acetate containing 2000 U/mL of neuraminidase (New England Biolabs P0720) and then blocked with 5% normal goat serum or BSA in Tris-buffered saline with 0.1% Tween-20 (TBST) for 1 h at room temperature. Slides were incubated with primary antibodies (5 μg/ml) or biotinylated SNA (20 μg/ml) (Vector Labs) ON at 4 °C. Next, slides were incubated with HRP-conjugated anti-rabbit, anti-mouse or streptavidin secondary antibodies (KPL Inc., Gaithersburg, MA) at room temperature for 1 h. Signals were visualized by incubating sections with Aminoethylcarbazole (AEC) substrate solution (Invitrogen), and cell nuclei were counterstained with hematoxylin (Invitrogen). Slides were imaged on ImageXpress Pico Automated Cell imaging system (Molecular Devices). Human tissue sections mounted on glass slides in 5-μ sections were obtained from US Biomax, Rockville, MD.

**Statistics and reproducibility**. The CFG glycan microarray experiments were performed and supervised by the staff at the core facility located at Emory University, and subsequently at Beth Israel Deaconess Medical Center (BIDMC). The core facility routinely performs quality control assays on the glycan microarrays using plant lectins with defined specificity, to ensure that the data collected from the microarrays are reproducible. This data are available upon request and at the NCFG website. The CFG glycan microarray contains over 600 unique glycan structures, each printed in replicates of six[62]. The data were analyzed by measuring the fluorescent intensity of a defined feature (printed glycan), and the fluorescent intensity surrounding the feature. The levels of local background intensity were subtracted from each glycan feature. Relative fluorescent units (RFUs) were calculated by taking the mean of the intensity in four of the six features, with the highest and the lowest values of the six replicates being eliminated from the analysis to avoid bias. Positive binders were evaluated based on the standard deviation of the mean, and the calculated coefficient of variance (%CV). Positive binders were considered if the mean RFU was above 100, and has a %CV under 30. All of the microarray data has been normalized to a scale from 0 to 1, to reflect the percent of positive binders per sample. Force graphs and Pearson r correlation coefficients were calculated using standard analytical methods used in the GLAD online software[42].

**Reporting summary**. Further information on research design is available in the Nature Research Reporting Summary linked to this article.

## Data availability

All raw microarray data is provided in the supplementary data files and individual monoclonal VLRBs will be made publicly available upon request.

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

## Acknowledgements

This work was funded by the National Institute of Health (P41GM103694, U01CA199882, and R01AI072435), and this project was supported by award number T32AI074492 from the National Institute of Allergy and Infectious Diseases, and the Independent Research Fund Denmark (7025-00083B).

## Author contributions

R.D.C. and M.D.C. are the primary PIs and provided mentorship, advice, and contributed to preparation of the paper. T.R.M. executed the majority of the experiments, conducted the data analysis, and prepared the paper. C.K.G. performed the I.H.C. staining. C.S.R. contributed to protein expression and experimental design. H.N. carried out the mouse experiments. J.H.M. contributed to the design and interpretation of the microarray results, writing and mentorship. B.R.H. contributed to the construction of the Y.S.D. libraries and mentorship. A.M.M. performed and analyzed many of the microarray experiments. R.F. and N.J.R. assisted in identifying new VLRB proteins and microarray analysis.

## Competing interests

The authors declare no competing interests.
