## [Peer Review File · Communications Biology]

Reviewers' comments:

Reviewer #1 (Remarks to the Author):

1. Manuscript background information

The manuscript is a well-conceived and elegant work about the development of multiple Smart Anti-Glycan Reagents (SAGR) using lampreys xeno-immunized with whole fixed cells (CHO), tissue homogenates (pig lungs), and human milk as immunogens. Briefly, after xeno-stimulation, cDNA from lampreys' lymphocytes are purified, cloned, expressed and then exposed on the surface of yeast as a library (YSD libraries) of immunoglobulins (named VLRB-Ig chimeras). An essential part of this library is composed by Ig chimeras targeting glycan moieties, called by the authors as SAGR, which can be used to study glycan expression. To document the robustness of this method, the authors characterized many novel SAGRs, which are useful for all types of glycan analyses, including immunohistochemistry, Western blots, and flow cytometry. This is an excellent manuscript with plenty of experimental work.

2. Comments for transmission to the authors

The authors addressed the experimental work accordingly with the objectives of the paper. The manuscript is very well-structured and easy to follow; readers will appreciate it. The authors claimed the use of Smart Anti-Glycan Reagents (SAGR) libraries for different application. The novelty of this work is the generation of this library. The authors generate the library by xeno-immunization using whole fixed cells, tissue homogenates, and human milk. Hence, an ample and diverse library of Ig targeting residues of human self-antigens, including glycan residues, is supposed to be represented. The applicability of SAGRs and the availability of YSD libraries may enhance future studies on glycan expression by providing sequenced, defined antibodies for a variety of research. For sure, the present work will be of maximum interest to the glycomic community but even to people from other fields interested in study the expression profile of glycans in health and disease. The methodology, in general, is elegant, with special mention to the enrichment strategy.

The novelty of the manuscript comes from the possibility of using SAGR to follow the expression of multiple glycans. They claim the capability of the lamprey to generate a diverse collection of anti-glycan antibodies and delineate the methodology for translating this response into a reagent that can be broadly used for traditional immunological research applications. Glycan expression is often studied using lectins or monoclonal antibodies. To my knowledge, lectins recognize multiple glycan structures. Hence their use is limited due to broad specificity. Monoclonal antibodies (MAb) are specific and very valid to test the expression of a single carbohydrate. But is not feasible to generate by classical development one single MAb for each residue of carbohydrate expressed by humans. However, my concern is about the use of lamprey instead of murine models for SAGR generation. I consider that similar methodology for lamprey may apply to murine models. Thus, the paper needs to delineate why to use this model instead of murine or any other vertebrate. Regarding design, there is a complete absence of statistical analysis in the manuscript, although they are necessary from design, selection of animal numbers to comparisons, correlations and PGA analysis.

The work is easy to follow, and the authors provide enough detail to reproduce all the experimental. Conclusions are originals.

Formatting the review

1. Brief summary of the manuscript

The present work is about a novel strategy that potentially can be used to discern the temporal expression of glycans in health and disease. Briefly, the authors xeno-immunized lampreys with whole fixed cells, tissue homogenates, and human milk. The cDNA from the lympho-B population, as a result of xeno-stimulation, was cloned, and expressed in yeast to generate a library of VLRB-Ig chimeras. Within this library, Ig chimeras targeting glycan moieties were named by the authors as SAGR. The authors study some of the novel SAGR, demonstrating its utility for discerning the temporal glycan

analysis by immunohistochemistry, Western blots, and flow cytometry. As an example, the authors showed a first proof of concept with novel sialic acid binding SAGRs that provides an innovative approach to accurately dissect the precise structure of sialylated glycans present in different tissues.

2. Overall impression of the work

This is an excellent manuscript with plenty of experimental work. The novelty of this work comes by the ample population of anti-glycan antibodies that may be produced by xeno-immunization due to the nature of the used immunogens: whole fixed cells, tissue homogenates, and human milk. Glycans are recognized as key players in human metabolism. However, in many cases, their function, expression, localization is largely unknown. The present work opens a new avenue to discern the temporal expression of carbohydrates by immunohistochemistry, Western blots and flow cytometry.

3. Specific comments, with recommendations for addressing each comment

Here are some recommendations to improve manuscript quality:

Major concerns:

1- murine vs lampreys.

a) Is it feasible to develop the same strategy in a murine model (mouse or rat)? To extract total RNA from leukos, generate cDNA libraries, etc..

b) Why do you select lamprey vs. murine model for xeno-immunization? More repertoire of anti-glycan antibodies? A unique repertoire in Lamprey? Technical constraints?

Line 244: "For both of these sets of immunization, the VLRB and IgM profiles were highly divergent for type AB erythrocytes (Fig. 2g, $r=0.2386$) and SIV particles (Fig. 2h, $r=0.1485$)",

Mouse is the most used model for antibodies generation. There are references describing that only the repertoire of natural-circulating anti-glycan antibodies (naïve adult animals) is formed by approx. 100 different glycan specificities (Ref 1, 2). Naïve lampreys do not have, according to Fig 1, natural anti-glycan antibodies.

Prior art in this field can justify the use of xeno-immunization in rodents to generate anti-glycan antibodies (Ref 3, 4).

In the manuscript: at least for IgM, after SIV particles immunization, mouse produced a higher diversity (glycan specificities, repertoire) and levels of anti-Glycan antibodies compare to Lamprey (Fig 2D- 2F).

c) In any case, the readers will appreciate in the introduction a more explicit justification for using lampreys instead of any other strategy.

There is information related to this issue in the discussion (Line 392) but better to clarify this point from the very beginning.

Line 247: "The results of these experiments suggests that the lamprey is a complementary alternative to a mouse model for generating anti-glycan binding reagents. While both the lamprey and mouse produced anti-glycan antibodies after immunization, the advantage of the lamprey system lies within the downstream processing of the reagents which is far simpler, less expensive, and more reliable than the creation of hybridomas, as well as providing permanent cDNA libraries encoding all lamprey antibodies induced"

Please, rephrase this affirmation considering other technologies.

Lamprey could be a complementary model to mouse but notice that repertoire (glycan diversity= specificities) and levels of anti-glycan antibodies is higher in mouse compare to the lamprey.

There are more strategies different to hybridomas for screening and selecting potential SAGR, e. g: Fab phage display. Hybridomas is obviously a time consuming and costly process.

Ref 1: Front Immunol. 2019 Mar 5;10:342. doi: 10.3389/fimmu.2019.00342.

Ref 2: Front Immunol. 2017 Nov 6;8:1449. doi: 10.3389/fimmu.2017.01449.

Ref 3: J Innate Immun. 2014;6(2):140-51. doi: 10.1159/000355305.

Ref 4: PLoS One. 2015 May 18;10(5):e0125472. doi: 10.1371/journal.pone.0125472.

2- Fig 2: murine versus lamprey systems

Line 229: "Mice generated only minor anti-glycan IgG response to the antigens within this time frame

(Fig. 2a-b)

a) Any explanation, it seems the immunization scheme may be related to this result? Animal age?

b) Please include in the legend of Fig 2 a brief description of immunization scheme. It is not clear when you've performed the bleeding and analysis after the xeno-immunization.

c) Did you evaluate IgG in lamprey, in Fig 2?

3- The screening and selection of SARS

First, the general strategy of screening and enrichment is very well summarized in Figure 7, readers will appreciate it. Thanks.

a) glycan array with 600 unique glycans

Could you please delineate the origin? source of the printed glycans? Purity?

One major issue here is to demonstrate that glycans are exposed in similar configuration than natural ones. Configuration should be the same as natural.

b) any explanation for no detecting glycan-binding for unsorted YSD library?? (Fig 3D). It is well described (Ref 5) the presence of lectins in the surface of yeast; thus it is expected some basal recognition in the PGA

Ref 5: Biotechnol Adv. 2011 Nov-Dec;29(6):726-31. doi: 10.1016/j.biotechadv.2011.06.002.

c) a general observation: many signals you consider positive in PGA are below 2000 RFU. PGA per se generates background, 500-1000-2000 RFU may be considered in many cases background of the experiment.

d) When you use direct lamprey plasma, it is more accurate to graphic the ratio between plasma before immunization and after immunization, to discard any background from the animal model. Did you consider it?

e) Figure 4 (enrichment strategy): why is so different direct PGA (a) and after enrichment (C and E).

- If you were capable to generate this SARS after immunization why many of them are not detectable in a, before enrichment

- Here, many positive signals that you are considering as positive in C and E are below 2000, this is almost PGA background.

- Is there any explanation for such decrease in the levels of BGO and sulfated SARS after enrichment (They pass from 20000 RFU to maximum 8000 RFU)

- Please, I would like to see these results using the same scale for a, c and e.

4- STATISTIC:

There is not any section dedicated to the statistic. This work requires from design to analysis a complete statistical analysis. For example, the authors used Pearson correlation but there is not any methodology in methods

a) There is no statistical justification in the number of animals used in every experiment.

b) There is a lot of data to analyze and compare and there is not a single statistical method described in the manuscript for such purposes

c) Additionally, PGA printing and analysis requires statistical. PGA generate a lot of raw data that must be analyzed using the correct statistical tool. There is not a description.

d) How many replicates of every glycan are in the slide of PGA (3, 4, 6?).

e) How is expressed the results obtained by PGA: RFU as mean \pm SD or median \pm MAD ??

Please, include all this information in a special section in methods.

Additional comments:

1- It is difficult to establish a comparison in Figure 1 cause different scales. Even the naïve animal has a maximum scale of 4000, which can be considered in these types of analysis as the background of the experiment. You may consider changing the scale of graphics to facilitate comparisons.

2- Line 202: "Naïve plasma lacked significant anti-glycan responses (Fig. 1h), Is there any reason for such very low levels (almost inexistent) for natural anti-glycan antibodies in the naïve animals? Is it because of a reduced diversity or representation of glycans in the array? Screening technology?? or physiology of lamprey?"

3- regarding PFA fixed cells:

PFA promotes protein cross-linking and can potentially mask antigen of interest. Did you consider using live cells instead of fixed for immunization?

4- There is according to Fig. 1h only one natural anti-glycan antibody present in the naïve lamprey. However, there is a huge variation in the level of this natural antibody comparing with the rest of lampreys. Please, explain a little bit this finding considering previous studies of repertoire of natural anti-glycan antibodies performed in other species.

5- Suggestion in Fig 1: better to detect also IgM since bound VLRBs were detected with anti-VLRB murine IgG (4C4)27 and Alexa-Fluor anti-mouse IgG secondary.

a) Is there any technical or logistic reason to discard IgM analysis in Fig 1? also considering that main analysis performed in Fig 2 is IgM.

6- Line 207: "Overall, each biological sample stimulated a relatively unique anti-glycan VLRB response".

Line 291: "As to be expected, the plasma obtained from lampreys immunized with type O libraries contained VLRBs primarily against the type 2 BGO and sulfated H-antigen glycans, and at first glance the profile appeared to be fairly uniform (Fig. 4a).

a) Did you find also differences (levels and repertoire) between animals (same immunogen) for the rest of biological samples used as immunogens? If so, please add this information, and if possible, include references?

7- Xenoinmunization and anti-glycan antibodies screening

a) Could you, please, compare the repertoire of anti-glycan antibodies generated by xenoinmunization (pig lungs) with previous papers (ref 3-4)?

8- Please, correct in the supplementary files (1-3) alpha and beta: $a/b = \alpha/\beta$.

9- Please, in the supplementary files (1-3), clarify units; median or mean? and deviation for every single glycan.

In Fig 1-4 results are expressed as average RFU, average means arithmetic mean? If so, please clarify why you prefer to use mean instead of median.

10- For glycan array analysis, please clarify cut-off for positive and the experiment background And also, number of replicates for each single glycan in the array.

11- line 395: "...presented here is complementary, but in many ways superior, to conventional approaches..

I would say just complementary.

12- Number of Lamprey animals, $n=3$, is it supported statistically speaking?

13- Murine immunizations

- Animal age: Six to eight-week-old BALB/cAnNHsd mice...

You are comparing Lampreys (2-4 years, I guess adults) with mice of a maximum of 8 weeks old. To my knowledge, the immune system of 6-8 weeks old mice is in the process of maturation. You can see what happen specifically with the repertoire of natural anti-glycan antibodies in mice (see figure 2). Front Immunol. 2019 Mar 5;10:342. doi: 10.3389/fimmu.2019.00342.

a) Did you consider using older animals?

14- Serum collection:

Line 502: "Blood was collected one day prior to immunization, and two weeks after the final injection. Total blood was allowed to clot, overnight at 4°C and centrifuged at 400 xg for 15 minutes.

a) clotting all night long? This is not advisable for analyzing the repertoire of anti-glycan antibodies by PGA. Although incubation is at 4°C, the long incubation period may allow unspecific interaction of antibodies with the clot, resulting in a lower level and diversity of antibodies in the analysis.

Suggestion: 45 min at 25°C is enough.

b) Additionally, did you process lamprey and mice serum the same way?

If you use this serum for evaluating the repertoire of anti-glycan antibodies after immunization, both lamprey and mice serum must be processed identically.

This is not clear in materials and methods.

Line 492: Lamprey plasma was collected and stored at 4°C and examined for positive VLRB titers

Line 502: "Blood was collected one day prior to immunization, and two weeks after the final injection. Total blood was allowed to clot, overnight at 4°C and centrifuged at 400 xg for 15 minutes.

15- If possible, please, add references:

Line 556: "The slides with the bound yeast were dried, scanned and analyzed according to the standard CFG protocol

16- Please justify why so long time for incubation:

Line 556: "Induced YSD libraries were incubated overnight, gently shaking at 4°C prior to washing and analyzing"

Reviewer #2 (Remarks to the Author):

In this manuscript, McKittrick et al describe an optimized workflow to generate antibody reagents for the selective recognition of mammalian glycans. This workflow builds on previous work (duly referenced by the authors) that combined lamprey antibody generation with yeast surface display to generate antibody libraries. A key contribution of this work is the design and selection of lamprey-derived Smart anti-glycan reagents for the recognition of mammalian carbohydrates, which have been challenging to generate in traditional antibody workflows. The current lack of reagents for glycan detection and analysis is crippling for glycobiologists, so I was absolutely delighted to read this manuscript. I think this approach could lower the bar for entry into glyco research, which should greatly accelerate progress in the field.

The authors acknowledge that the method described is limited by the sample availability on the glycan arrays. As Prof. Cummings is the primary provider of glycan array screenings through the NCFG, it would be nice if the authors were able to comment on any progress, if any, to create shotgun glycan analysis platforms that are described in the discussion. Additionally, while this is presented as a method that others could adopt to create their own reagents, Prof. Cummings is probably the best equipped to perform this antibody generation and characterization. Will an antibody generation service be added to the NCFG, or will the antibodies generated in this work be available through a central resource? If so, it would be helpful to make this clear in the discussion.

Overall, I am excited by this work and I think the Smart anti-glycan reagents will be enormously useful in filling a huge gap in glycobiology.

Reviewer #3 (Remarks to the Author):

This is a manuscript on an exciting new tool development program focusing upon the creation of glycan detection reagents. Historically, most detection on cells, Western blots, IHC, and other methods typically employing antibodies have been the domain of plant lectins. Some of the plant lectins are excellent. Others are notoriously problematic in terms of specificity and reliability. There is a potent need for better reagents, and this is where the significance of the work described herein is found.

The manuscript is very well written, and the reagents are exciting. The extensive use of glycan arrays is both a strength and weakness, as pointed out in the Discussion. Only those with the ability to work with glycan arrays could hope to repeat this work; however, the point is not to repeat it, but to use the reagents as tools. In fact, the specificity shown in Figure 6 is rather striking and highlights the need for these reagents above and beyond even those plant lectins that are very reliable.

Finally, these detection reagents should appeal to a very broad range of investigators, making this manuscript quite broadly impactful.

Only a few comments should be addressed in a revision:

-In the initial immunization studies (Fig 1), it would have been nice to have a detailed glycomic analysis of each cell or tissue being used so that the relative coverage of lamprey response could be judged. Put another way, how abundant does the glycan need to be in order to get a robust response in the lamprey system? This was known for some of the analyses, but the Tn4 B cells and pig lung data are less clear in terms of what should be expected.

-A point of concern in terms of usability of these reagents is found in lines 343-355. More specifically, it is clear that the "type of linker that couples the sugar to the slide" and other "presentation" details significantly impacts the binding of a given reagent. It may turn out that the glycan ligand for one of these reagents is not recognized within a biological sample (due to presentation differences), leading to the potential for false negatives. While this should not impede publication, suggestions for mitigating this should be included in the Discussion.

-Minor: The statement on line 221-222 about self-tolerance and processing should have the appropriate citations.

-Minor: The authors discuss the advantages of the lamprey system over mIgM on lines 248-252. Isn't one of the other advantages that the lamprey molecules are more specific (less cross-reactive)?

Response to Reviewer Comments- McKitrick et al, *CommsBio- COMMSBIO-19-0872-T*

Referee expertise:

Referee #1: Glycan-specific antibodies

Referee #2: Antibody production

Referee #3: Immunology, antibody production

Reviewers' comments:

Reviewer #1 (Remarks to the Author):

1. Manuscript background information

The manuscript is a well-conceived and elegant work about the development of multiple Smart Anti-Glycan Reagents (SAGR) using lampreys xeno-immunized with whole fixed cells (CHO), tissue homogenates (pig lungs), and human milk as immunogens. Briefly, after xeno-stimulation, cDNA from lampreys' lymphocytes are purified, cloned, expressed and then exposed on the surface of yeast as a library (YSD libraries) of immunoglobulins (named VLRB-Ig chimeras). An essential part of this library is composed by Ig chimeras targeting glycan moieties, called by the authors as SAGR, which can be used to study glycan expression. To document the robustness of this method, the authors characterized many novel SAGRs, which are useful for all types of glycan analyses, including immunohistochemistry, Western blots, and flow cytometry. **This is an excellent manuscript with plenty of experimental work.**

***Response of Authors:** We thank the reviewer for their thorough review of our work, and we have addressed each comment below with a point-by-point response and have indicated where the changes are located in the manuscript.*

2. Comments for transmission to the authors

The authors addressed the experimental work accordingly with the objectives of the paper. The manuscript is very well-structured and easy to follow; readers will appreciate it. The authors claimed the use of Smart Anti-Glycan Reagents (SAGR) libraries for different application. The novelty of this work is the generation of this library. The authors generate the library by xeno-immunization using whole fixed cells, tissue homogenates, and human milk. Hence, an ample and diverse library of Ig targeting residues of human self-antigens, including glycan residues, is supposed to be represented. The applicability of SAGRs and the availability of YSD libraries may enhance future studies on glycan expression by providing sequenced, defined antibodies for a variety of research. For sure, the present work will be of maximum interest to the glycomic community but even to people from other fields interested in study the expression profile of glycans in health and disease. The methodology, in general, is elegant, with special mention to the enrichment strategy.

The novelty of the manuscript comes from the possibility of using SAGR to follow the expression of multiple glycans. They claim the capability of the lamprey to generate a diverse collection of anti-glycan

antibodies and delineate the methodology for translating this response into a reagent that can be broadly used for traditional immunological research applications. Glycan expression is often studied using lectins or monoclonal antibodies. To my knowledge, lectins recognize multiple glycan structures. Hence their use is limited due to broad specificity. Monoclonal antibodies (MAb) are specific and very valid to test the expression of a single carbohydrate. But is not feasible to generate by classical development one single MAb for each residue of carbohydrate expressed by humans. However, my concern is about the use of lamprey instead of murine models for SAGR generation. I consider that similar methodology for lamprey may apply to murine models. Thus, the paper needs to delineate why to use this model instead of murine or any other vertebrate. Regarding design, there is a complete absence of statistical analysis in the manuscript, although they are necessary from design, selection of animal numbers to comparisons, correlations and PGA analysis. The work is easy to follow, and the authors provide enough detail to reproduce all the experimental. Conclusions are originals.

Response of Authors: *We appreciate the thoughts of the reviewer in regard to the lamprey system versus the murine system. The murine glycome is very similar to the human glycome, and indeed to all mammals. Thus, it has been technically difficult over the years in the field to develop robust antibodies to all kinds of mammalian glycans. We surmise that the glycome of the lamprey, due to the hundreds of millions of years of evolution that has separated the jawed versus jawless vertebrates, might be relatively different enough to allow generation of larger repertoires of anti-carbohydrate antibodies to mammalian glycans. This is part of the rationale for our undertaking. To our knowledge, as we will discuss more below, there has been no similar approach that has been taken for identifying anti-carbohydrate antibodies using libraries generated from cDNAs of murine lymphocytes after immunization. We would think the reviewer would agree, that if such a technology was available, it would be useful also, but that is beyond the scope of our interest here. We have now revised our manuscript to indicate this rationale in the introduction and again in the discussion.*

Formatting the review

1. Brief summary of the manuscript

The present work is about a novel strategy that potentially can be used to discern the temporal expression of glycans in health and disease. Briefly, the authors xeno-immunized lampreys with whole fixed cells, tissue homogenates, and human milk. The cDNA from the lympho-B population, as a result of xeno-stimulation, was cloned, and expressed in yeast to generate a library of VLRB-Ig chimeras. Within this library, Ig chimeras targeting glycan moieties were named by the authors as SAGR. The authors study some of the novel SAGR, demonstrating its utility for discerning the temporal glycan analysis by immunohistochemistry, Western blots, and flow cytometry. As an example, the authors showed a first proof of concept with novel sialic acid binding SAGRs that provides an innovative approach to accurately dissect the precise structure of sialylated glycans present in different tissues.

2. Overall impression of the work

This is an excellent manuscript with plenty of experimental work. The novelty of this work comes by the ample population of anti-glycan antibodies that may be produced by xeno-

immunization due to the nature of the used immunogens: whole fixed cells, tissue homogenates, and human milk. Glycans are recognized as key players in human metabolism. However, in many cases, their function, expression, localization is largely unknown. The present work opens a new avenue to discern the temporal expression of carbohydrates by immunohistochemistry, Western blots and flow cytometry.

Response of Authors: *We appreciate the positive comments from this reviewer.*

3. Specific comments, with recommendations for addressing each comment

Here are some recommendations to improve manuscript quality:

Major concerns:

1- murine vs lampreys.

a) Is it feasible to develop the same strategy in a murine model (mouse or rat)? To extract total RNA from leukos, generate cDNA libraries, etc..

1a Response of Authors: *We find it difficult to comment on this, as our work is independent of studies in rodents, but as we mentioned above, this issue of relative differences between lamprey and murine glycomes may help make the lamprey more amenable to obtaining anti-mammalian glycan antibodies. To our knowledge no laboratory has taken a similar approach to ours using murine lymphocytes. While there are considerable technical challenges associated with this methodology, the protocols have been well documented in the literature for constructing non-immunized and immunized immunoglobulin libraries (Miller et al 2008, Chao et al 2006). These references are now included in our revised manuscript.*

b) Why do you select lamprey vs. murine model for xeno-immunization? More repertoire of anti-glycan antibodies? A unique repertoire in Lamprey? Technical constraints?

Line 244: “For both of these sets of immunization, the VLRB and IgM profiles were highly divergent for type AB erythrocytes (Fig. 2g, $r=0.2386$) and SIV particles (Fig. 2h, $r=0.1485$)”,

Mouse is the most used model for antibodies generation. There are references describing that only the repertoire of natural-circulating anti-glycan antibodies (naïve adult animals) is formed by approx. 100 different glycan specificities (Ref 1, 2). Naïve lampreys do not have, according to Fig 1, natural anti-glycan antibodies.

Prior art in this field can justify the use of xeno-immunization in rodents to generate anti-glycan antibodies (Ref 3, 4).

In the manuscript: at least for IgM, after SIV particles immunization, mouse produced a higher diversity (glycan specificities, repertoire) and levels of anti-Glycan antibodies compare to Lamprey (Fig 2D- 2F).

1b Response to reviewer: *We have now made it clearer that the purpose of our developments are to explore the anti-glycan repertoire induced by xeno-immunization in the lamprey, and develop a high*

throughput methodology to isolate and utilize these unique proteins for traditional research applications. While the immune system of the lamprey has been studied for over 50 years, the breadth and diversity of anti-glycan VLRBs generated after immunization has never been explored until this study. The data suggests that the lampreys are able to mount a diverse anti-glycan VLRB response after immunization with different antigens, recognizing a myriad of mammalian glycan structures. The simplicity of the single chain VLRBs and ease of expression in eukaryotic cells is a significant technical advantage, and provides a versatility to the reagents that is not often afforded in other model systems.

While we agree that the murine model is the most widely used system for the generation of monoclonal antibodies, and we do not suggest replacing such a well-known system, we are excited about the novel development in our studies of anti-glycan libraries of antibodies using the lamprey and yeast surface display technology. To our knowledge this is not replicable in murine systems, as the lamprey is a single-chain antibody. Thus, we envision that the lamprey system offers an alternative and complementary system to typical murine antibody development. As pointed out by reviewer #1, the antibody response in the mouse and lamprey were quite divergent from each other, meaning that the IgM antibodies generated in the mouse immunizations were recognizing a different class of glycans than that of the lamprey VLRB proteins using the same immunogen. We have modified Figure 2, and included an analysis of the mouse serum prior to immunization. This discovery that we have included demonstrates that the mice used in this experiment do contain a broad repertoire of anti-glycan IgMs as pointed out by the reviewer, and which we have cited as suggested by the reviewer. However, our observation that the lamprey are generating unique anti-glycan VLRBs that are non-overlapping from the murine model (either pre-existing or induced by immunization) is still valid and provides further evidence for the need for additional platforms to generate these reagents. We feel strongly that multiple platforms to generate reagents, now including the lamprey system, may be useful in deciphering the expression of the mammalian glycome.

While naïve lamprey larvae do not express an appreciable amount of circulating anti-glycan VLRBs, there is ample evidence in the literature that the lamprey possess the genetic sequences to generate the VLRB anti-glycan repertoire. Panzer and colleagues discovered a wide variety of anti-glycan binding reagents from a naïve VLRB yeast surface display library created from 100 animals (reference numbers 11,12). Given these observations, we set out to discover if the lampreys would mount a specific response towards a given antigen and increase the amount of circulating VLRBs present in the plasma.

c) In any case, the readers will appreciate in the introduction a more explicit justification for using lampreys instead of any other strategy.

Ic Response of Authors: *We have included an explicit justification of this approach in a new paragraph that begins on page 10 on line 263.*

d) There is information related to this issue in the discussion (Line 392) but better to clarify this point from the very beginning.

Line 247: “The results of these experiments suggests that the lamprey is a complementary alternative to a

mouse model for generating anti-glycan binding reagents. While both the lamprey and mouse produced anti-glycan antibodies after immunization, the advantage of the lamprey system lies within the downstream processing of the reagents which is far simpler, less expensive, and more reliable than the creation of hybridomas, as well as providing permanent cDNA libraries encoding all lamprey antibodies induced”

Please, rephrase this affirmation considering other technologies.

“Lamprey could be a complementary model to mouse but notice that repertoire (glycan diversity= specificities) and levels of anti-glycan antibodies is higher in mouse compare to the lamprey.”

There are more strategies different to hybridomas for screening and selecting potential SAGR, e. g: Fab phage display. Hybridomas is obviously a time consuming and costly process.

Ref 1: Front Immunol. 2019 Mar 5;10:342. doi: 10.3389/fimmu.2019.00342.

Ref 2: Front Immunol. 2017 Nov 6;8:1449. doi: 10.3389/fimmu.2017.01449.

Ref 3: J Innate Immun. 2014;6(2):140-51. doi: 10.1159/000355305.

Ref 4: PLoS One. 2015 May 18;10(5):e0125472. doi: 10.1371/journal.pone.0125472.

Id Response of Authors: *As we stated in the previous comment, our intent is to identify specificities of anti-carbohydrate antibodies, with the hope that many will be non-overlapping and distinct, thus the lamprey are making a suite of VLRBs against glycans that are not bound by IgM. However, the reviewer brings up a really good point regarding the levels of anti-glycan antibodies. The microarray assays are not quantitative, the serum or plasma is screened using different secondary reagents (IgM v VLRB) and thus cannot be directly compared with regard to intensity. The intention of the microarray assays is to compare the pattern of binding, not the RFU values directly. Additionally, we did not perform any independent assay to quantitatively measure antibody titers between the mouse and lamprey, and the microarray should not be interpreted as such. In order not to confuse the readers, all of the microarray data will be normalized to a scale of 0-100, to reflect the percent of positive binding.*

As per the reviewer’s request, we have included in our revised text beginning on line 261 the discussion of alternate display technologies.

2- Fig 2: murine versus lamprey systems

Line 229: “Mice generated only minor anti-glycan IgG response to the antigens within this time frame (Fig. 2a-b)

a) Any explanation, it seems the immunization scheme may be related to this result? Animal age?

2a Response of Authors: *We did not spend a lot of time exploring the murine IgG system, as our study is focused on developing the lamprey model. It is certainly possible that the immunization scheme may have resulted in a diminished IgG response, which could arise for a number of reasons, including age of the animals, time series of the immunizations (only 6 weeks) and lack of adjuvant. Again, the purpose of these*

experiments was to examine the specificity of the antibodies generated in a murine model if we used the same antigens for immunization as used for the lamprey.

b) Please include in the legend of Fig 2 a brief description of immunization scheme. It is not clear when you've performed the bleeding and analysis after the xeno-immunization.

2b Response of Authors: *The serum collected from the mouse was analyzed 2 weeks after the last immunization, similar to what was done in the lamprey. The Fig. 2 figure legend has been modified accordingly.*

c) Did you evaluate IgG in lamprey, in Fig 2?

2c Response of Authors: *Again, to clarify to the reviewer, lampreys do not produce immunoglobulins as antigen receptors, because immunoglobulins are only produced in mammalian systems. Lampreys generate VLRB as antigen receptors, which we screened as shown in Fig. 2.*

3- The screening and selection of SAGRS

First, the general strategy of screening and enrichment is very well summarized in Figure 7, readers will appreciate it. Thanks.

a) glycan array with 600 unique glycans

Could you please delineate the origin? source of the printed glycans? Purity?

One major issue here is to demonstrate that glycans are exposed in similar configuration than natural ones. Configuration should be the same as natural.

3a Response of Authors: *The glycan microarray produced by the Consortium for Functional Glycomics (CFG) was the result of a concerted effort of many researchers at several institutions, and ultimately produced one of the most comprehensive glycan microarrays available (over 600 unique structures). The glycans were chemically and enzymatically synthesized, and coupled to a distinct class of linkers that enabled the glycans to be printed onto the glass surface (Blixt et al 2004). Considerable effort was taken to ensure that the glycans were pure and quality controlled. The CFG operated as a NIH funded service for over a decade, and the technicians at the CFG core facility previously located at Emory University ran many of the microarray analyses presented within this manuscript. A full delineation of the sequence of the glycans can be found in the supplementary files, as well as the website functionalglycomics.org.*

The glycan microarrays are not designed to 'mimic' the natural display of all glycans as in normal cells and tissues. Rather, the microarrays display glycans and their composite 'epitopes' for identifying glycan recognition by glycan-binding proteins and antibodies. The microarrays are essential in our work for helping to define the epitopes recognized by antibodies, as well as for screening yeast surface displayed antibodies.

b) any explanation for no detecting glycan-binding for unsorted YSD library?? (Fig 3D). It is well

described (Ref 5) the presence of lectins in the surface of yeast; thus it is expected some basal recognition in the PGA

Ref 5: Biotechnol Adv. 2011 Nov-Dec;29(6):726-31. doi: 10.1016/j.biotechadv.2011.06.002.

3b Response of Authors: *As we briefly mentioned on line 301 in the revised manuscript, the unsorted library only had a few bound yeast colonies on the array that could be viewed macroscopically. There were not a sufficient enough number of colonies bound to be detected by the microarray scanner. As far as the detection of lectins expressed on the cell surface of yeast, it has been documented in the literature that saccharomyces expresses lectins that bind to mannose-containing glycans, but we have no evidence that the yeast used in our studies express functional lectins or the glycans that may be recognized by a yeast lectin might not be on the microarrays, as yeast normally do not bind to our glycan microarrays. It is also possible that the induction and over expression of the VLRB protein on the cell surface of yeast may mask the fungal lectin. We have referenced the paper suggested by the reviewer (48) on line 295.*

c) a general observation: many signals you consider positive in PGA are below 2000 RFU. PGA perce generates background, 500-1000-2000 RFU may be considered in many cases background of the experiment.

3c Response of Authors: *We must respectfully disagree with the interpretation of the reviewer. The analysis of the glycan microarray takes into account any background present on the slide such as auto-fluorescence and dust particles that could obscure the binding pattern, and is subtracted from the analysis. Background on different glycan arrays varies according to the substrate, and the assay conditions. We routinely observe binding to the glycan arrays that are around 5-100 RFUs, so we don't consider any binding below 100. RFUs quantified at 1000-2000 may be low signal, but are real binding patterns that are reproducible. Low signal is a reflection of low affinity, or low abundance of the protein present in the sample, and we do not consider it background noise.*

d) When you use direct lamprey plasma, it is more accurate to graphic the ratio between plasma before immunization and after immunization, to discard any background from the animal model. Did you consider it?

3d Response of Authors: *It is not possible to collect blood from animals prior to immunization, the larvae are too small and we cannot extract sufficient amount of blood. These assays were performed on blood that was collected after exsanguination of the animal.*

e) Figure 4 (enrichment strategy): why is so different direct PGA (a) and after enrichment (C and E).

3e Response of Authors: *Figure 4a is the lamprey serum screened after immunization, whereas the figure C and E are yeast bound to the array. Proteins are always going to give a brighter signal than the yeast bound to the arrays, given the heterogenous expression pattern of the antibodies on the cell surface of the yeast. Due to this fact, it is not possible to make a direct comparison between the signal intensities of the two arrays and the graphs will be transformed to a scale from 0-1 in order to not confuse the readers (see above).*

f) If you were capable to generate this SARS after immunization why many of them are not detectable in a, before enrichment

3f Response of Authors: *It is most likely that many of the VLRBs generated after immunization are of low abundance, and possibly lower affinity. The results of the immunization demonstrate that antibodies against the type II H-antigen were the predominate response to type O erythrocytes. However, it is clear from 4C and E, that after enrichment of the YSD library, there are many more anti-glycan VLRBs present in the library than what is shown by the plasma.*

g) Here, many positive signals that you are considering as positive in C and E are below 2000, this is almost PGA background.

3g Response of Authors: *With all due respect to the reviewer, we disagree with this interpretation; this is not background, and see our point above. These are yeast colonies bound to the array, and the signal intensity cannot be directly compared to any other glycan array analysis, especially not proteins.*

h) Is there any explanation for such decrease in the levels of BGO and sulfated SARS after enrichment (They pass from 20000 RFU to maximum 8000 RFU)

3h Response of Authors: *To reiterate here as above, these are yeast colonies bound to the array, and the signal intensity cannot be directly compared to any other glycan array analysis, especially not proteins.*

i) Please, I would like to see these results using the same scale for a, c and e.

3i Response of Authors: *The arrays cannot be adjusted to the same scale, as it is not appropriate to directly compare the intensities. However, they can be transformed to a scale of 0-1 in order not to confuse the readers. We cannot stress this enough, the point is not the intensity but the patterns of binding.*

4- STATISTIC:

There is not any section dedicated to the statistic. This work requires from design to analysis a complete statistical analysis. For example, the authors used Pearson correlation but there is not any methodology in methods

a) There is no statistical justification in the number of animals used in every experiment.

4a Response of Authors: *As lamprey are not bred in captivity, and exist as larvae for 2-8 years, we are limited to using animals that are caught in the wild. Unfortunately, this implies that each individual has a unique genetic background and there will be inherent variation in the antibody response and has been observed in several experiments in the Cooper laboratory over the years. Thus, our standard protocol is to immunize 3 lamprey, determine which animal had the highest antibody titer against the given antigen and screen on the glycan microarrays, which is the standard method in the Cooper lab (#27). This was done using a variety of different methods, and is delineated on page 20-21 in the methods section. We did not screen all of the individuals on the glycan microarrays for several reasons, one being they are expensive and a limited resource, and the point of the manuscript was not to compare the population level*

response of lampreys to glycan targets but to assess the repertoire generated by animals which had been determined to have a positive titer and develop a methodology to isolate the anti-glycan antibodies.

b) There is a lot of data to analyze and compare and there is not a single statistical method described in the manuscript for such purposes

4b Response of Authors: *More detailed descriptions in this regard have been added to the methods section: Analysis of lamprey serum and VLRB protein specificity on the Consortium for Functional Glycomics (CFG) defined glycan microarray (Lines 556-568).*

c) Additionally, PGA printing and analysis requires statistical. PGA generate a lot of raw data that must be analyzed using the correct statistical tool. There is not a description.

4c Response of Authors: *More detailed descriptions in this regard has been added to the methods section: Analysis of lamprey serum and VLRB protein specificity on the Consortium for Functional Glycomics (CFG) defined glycan microarray (Lines 556-568).*

d) How many replicates of every glycan are in the slide of PGA (3, 4, 6?).

4d Response of Authors: *We use six replicates for each glycan on the microarrays. More detailed descriptions in this regard has been added to the methods section: Analysis of lamprey serum and VLRB protein specificity on the Consortium for Functional Glycomics (CFG) defined glycan microarray (Lines 556-568).*

e) How is expressed the results obtained by PGA: RFU as mean \pm SD or median \pm MAD ??

4e Response of Authors: *The RFU is expressed as the mean \pm SD. More detailed descriptions in this regard has been added to the methods section: Analysis of lamprey serum and VLRB protein specificity on the Consortium for Functional Glycomics (CFG) defined glycan microarray (Lines 556-568).*

Please, include all this information in a special section in methods.

Additional comments:

1- It is difficult to establish a comparison in Figure 1 cause different scales. Even the naïve animal has a maximum scale of 4000, which can be considered in these types of analysis as the background of the experiment. You may consider changing the scale of graphics to facilitate comparisons.

1 Response of Authors: *To address this concern of the reviewer, the scale will be normalized to 0-1, as direct comparisons is something we would like the readers to avoid. We are emphasizing the pattern of binding, not the intensity.*

2- Line 202: “Naïve plasma lacked significant anti-glycan responses (Fig. 1h),

Is there any reason for such very low levels (almost inexistent) for natural anti-glycan antibodies in the naïve animals? Is it because of a reduced diversity or representation of glycans in the array? Screening technology?? or physiology of lamprey?

2 Response of Authors: *It is unknown why naïve plasma does not contain significant anti-glycan responses, as this has never been examined before this manuscript and was not the primary focus. It is possible that the lamprey do have some circulating anti-glycan VLRBs, but perhaps not against the mammalian glycans represented on this array. The question we were addressing is whether after immunization, do the lampreys mount an anti-glycan response, and our results are affirmative in that regard.*

3- regarding PFA fixed cells:

PFA promotes protein cross-linking and can potentially mask antigen of interest. Did you consider using live cells instead of fixed for immunization?

3 Response of Authors: *When we are using lamprey to make antibodies against protein antigens, it is preferable to use live cells. However, for carbohydrate antigens this is less of a concern.*

4- There is according to Fig. 1h only one natural anti-glycan antibody present in the naïve lamprey. However, there is a huge variation in the level of this natural antibody comparing with the rest of lampreys. Please, explain a little bit this finding considering previous studies of repertoire of natural anti-glycan antibodies performed in other species.

4 Response of Authors: *Any positive signal with such high error bars can be considered as noise, and now is better explained in the methods section outlining the glycan microarray analysis.*

5- Suggestion in Fig 1: better to detect also IgM since bound VLRBs were detected with anti-VLRB murine IgG (4C4)27 and Alexa-Fluor anti-mouse IgG secondary.

a) Is there any technical or logistic reason to discard IgM analysis in Fig 1? also considering that main analysis performed in Fig 2 is IgM.

5 Response of Authors: *As indicated above, lamprey do not make immunoglobulins as antigen receptors, they only make VLRBs. We used a mouse monoclonal antibody (4C4) that specifically recognizes the secreted VLRB proteins found in lamprey plasma, which is then detected with fluorescent anti-mouse IgG.*

6- Line 207: “Overall, each biological sample stimulated a relatively unique anti-glycan VLRB response”.

Line 291: “As to be expected, the plasma obtained from lampreys immunized with type O libraries contained VLRBs primarily against the type 2 BGO and sulfated H-antigen glycans, and at first glance the profile appeared to be fairly uniform (Fig. 4a).

Did you find also differences (levels and repertoire) between animals (same immunogen) for the rest of biological samples used as immunogens? If so, please add this information, and if possible, include references?

6 Response of Authors: *We did not screen all of the individuals on the glycan microarrays that were immunized. We used the criteria outlined in the methods to determine which animal had the highest antigen specific titer, to screen and construct the YSD libraries with, and move forward with isolating the monoclonal antibodies.*

7- Xenoinmunization and anti-glycan antibodies screening

a) Could you, please, compare the repertoire of anti-glycan antibodies generated by xeno-immunization (pig lungs) with previous papers (ref 3-4)?

7 Response of Authors: *The papers suggested by the reviewer include immunizations with pig blood in hamsters and rats. We immunized with pig lung homogenate, and there is no reason to predict that the anti-glycan responses would be comparable between these two distinct cell types. We did observe VLRB binding to many structures found within the pig lung, as referenced by lines 164-169.*

8- Please, correct in the supplementary files (1-3) alpha and beta: $a/b = \alpha/\beta$.

8 Response of Authors: *The glycan sequences are listed with an a/b intentionally, so that is compatible with several online databases and data analysis software our group is developing as the special characters can be incompatible with some platforms. We will not be able to make that change requested, but we have indicated in the supplementary files that $a/b = \alpha/\beta$.*

9- Please, in the supplementary files (1-3), clarify units; median or mean? and deviation for every single glycan.

In Fig 1-4 results are expressed as average RFU, average means arithmetic mean? If so, please clarify why you prefer to use mean instead of median.

9 Response of Authors: *The average RFU is indicative of the mean fluorescence of 4 glycan spots. The arrays are printed in replicates of 6, but by convention we omit the highest and lowest values from the analysis to minimize outliers. We use the mean average instead of the median due to the variation that could occur from using a contact printer to generate the glycan microarrays. Inherently, there is pin to pin variation as the printing pin comes into contact with the surface. Thus, the mean RFU is a more robust measurement for this type of analysis.*

10- For glycan array analysis, please clarify cut-off for positive and the experiment background And also, number of replicates for each single glycan in the array.

10 Response of Authors: *More detailed descriptions in this regard has been added to the methods section: Analysis of lamprey serum and VLRB protein specificity on the Consortium for Functional Glycomics (CFG) defined glycan microarray (Lines 556-568).*

11- line 395: ..”presented here is complementary, but in many ways superior, to conventional approaches. I would say just complementary.

11 Response of Authors: *We clarified this statement above in a previous comment.*

12- Number of Lamprey animals, n=3, is it supported statistically speaking?

12 Response of Authors: *We clarified this statement above in a previous comment.*

13- Murine immunizations

- Animal age: Six to eight-week-old BALB/cAnNHsd mice.

You are comparing Lampreys (2-4 years, I guess adults) with mice of a maximum of 8 weeks old. To my knowledge, the immune system of 6-8 weeks old mice is in the process of maturation. You can see what happen specifically with the repertoire of natural anti-glycan antibodies in mice (see figure 2). Front Immunol. 2019 Mar 5;10:342. doi: 10.3389/fimmu.2019.00342.

Did you consider using older animals?

Response to reviewer: *The lamprey are in the larvae stage, however their age is unknown, as they are wild animals.*

14- Serum collection:

Line 502: "Blood was collected one day prior to immunization, and two weeks after the final injection. Total blood was allowed to clot, overnight at 4°C and centrifuged at 400 xg for 15 minutes.

a) clotting all night long? This is not advisable for analyzing the repertoire of anti-glycan antibodies by PGA. Although incubation is at 4°C, the long incubation period may allow unspecific interaction of antibodies with the clot, resulting in a lower level and diversity of antibodies in the analysis.

Suggestion: 45 min at 25°C is enough.

14 Response to reviewer: *We have developed this as our standard procedure, which has been perfected over a period of several years by trial and error using the lamprey blood. Also, see our response below to Point 15.*

15- Additionally, did you process lamprey and mice serum the same way?

If you use this serum for evaluating the repertoire of anti-glycan antibodies after immunization, both lamprey and mice serum must be processed identically.

This is not clear in materials and methods.

Line 492: Lamprey plasma was collected and stored at 4°C and examined for positive VLRB titers

Line 502: "Blood was collected one day prior to immunization, and two weeks after the final injection. Total blood was allowed to clot, overnight at 4°C and centrifuged at 400 xg for 15 minutes.

15 Response of Authors: *No, the lamprey and mice plasma were collected using different methodology. Lamprey require a minimum of 500 ul of 0.67x PBS with 30 mM EDTA in the collection tube in order to recover the lymphocytes for downstream processing, due to the presence of strong clotting factors in the plasma. It wasn't necessary to collect the lymphocytes from the murine samples, so we simply isolated the serum.*

16 - If possible, please, add references:

Line 556: "The slides with the bound yeast were dried, scanned and analyzed according to the standard CFG protocol

16 Response of Authors: *More detailed descriptions in this regard has been added to the methods*

section: Analysis of lamprey serum and VLRB protein specificity on the Consortium for Functional Glycomics (CFG) defined glycan microarray ((Lines 556-568).

17- Please justify why so long time for incubation:

Line 556: “Induced YSD libraries were incubated overnight, gently shaking at 4°C prior to washing and analyzing”

17 Response of Authors: *The yeast surface display library contains a large repertoire of VLRB antibodies, to both protein and glycan antigens. The long incubation time was required to isolate rare clones that could bind to the array, so as not to skew the library towards only the most predominate clones.*

Reviewer #2 (Remarks to the Author):

In this manuscript, McKittrick et al describe an optimized workflow to generate antibody reagents for the selective recognition of mammalian glycans. This workflow builds on previous work (duly referenced by the authors) that combined lamprey antibody generation with yeast surface display to generate antibody libraries. A key contribution of this work is the design and selection of lamprey-derived Smart anti-glycan reagents for the recognition of mammalian carbohydrates, which have been challenging to generate in traditional antibody workflows. The current lack of reagents for glycan detection and analysis is crippling for glycobiologists, so I was absolutely delighted to read this manuscript. I think this approach could lower the bar for entry into glyco research, which should greatly accelerate progress in the field.

Response of Authors: *We thank the reviewer for these positive comments on our manuscript, and we hope that our revisions strengthen the manuscript for publication.*

The authors acknowledge that the method described is limited by the sample availability on the glycan arrays. As Prof. Cummings is the primary provider of glycan array screenings through the NCFG, it would be nice if the authors were able **to comment on any progress, if any, to create shotgun glycan analysis platforms that are described in the discussion.**

Response of Authors: *We thank the reviewer for this comment. To clarify, we have generated numerous shotgun glycan microarrays to date- including from human and pig lung, and from human milk. These arrays are available to investigators upon request. As part of our ongoing research plans, we will create new shotgun glycan arrays in combination with lamprey immunizations in order to create libraries of tissue specific antibodies and glycans for further analysis.*

Additionally, while this is presented as a method that others could adopt to create their own reagents, Prof. Cummings is probably the best equipped to perform this antibody generation and characterization. **Will an antibody generation service be added to the NCFG, or will the antibodies generated in this work be available through a central resource? If so, it would be helpful to make this clear in the discussion.**

Response of Authors: *As part of our center, we will be making the lamprey antibodies and their sequences available to the scientific community.*

Overall, I am excited by this work and I think the Smart anti-glycan reagents will be enormously useful in filling a huge gap in glycobiology.

Response of Authors: We thank the reviewer for their positive comments.

Reviewer #3 (Remarks to the Author):

This is a manuscript on an exciting new tool development program focusing upon the creation of glycan detection reagents. Historically, most detection on cells, Western blots, IHC, and other methods typically employing antibodies have been the domain of plant lectins. Some of the plant lectins are excellent. Others are notoriously problematic in terms of specificity and reliability. **There is a potent need for better reagents, and this is where the significance of the work described herein is found.**

The manuscript is very well written, and the reagents are exciting. The extensive use of glycan arrays is both a strength and weakness, as pointed out in the Discussion. Only those with the ability to work with glycan arrays could hope to repeat this work; however, the point is not to repeat it, **but to use the reagents as tools.** In fact, the specificity shown in Figure 6 is rather striking and highlights the need for these reagents above and beyond even those plant lectins that are very reliable.

Response of Authors: As part of our center, we will be making the lamprey antibodies and their sequences available to the scientific community, as they can be used in a number of common laboratory techniques.

Finally, these detection reagents should appeal to a very broad range of investigators, making this manuscript quite broadly impactful.

Only a few comments should be addressed in a revision:

-In the initial immunization studies (Fig 1), it would have been nice to have a **detailed glycomic analysis** of each cell or tissue being used so that the relative coverage of lamprey response could be judged. **Put another way, how abundant does the glycan need to be in order to get a robust response in the lamprey system?** This was known for some of the analyses, but the Tn4 B cells and pig lung data are less clear in terms of what should be expected.

Response of Authors: We thank the reviewer for these comments. In order to provide more information on this point we will strengthen the discussion of our previous work describing the glycans found in pig and human lung, as well human milk. Unfortunately, we don't have any data that would inform us as to how abundant the glycan needs to be in order to get a robust response in the lamprey. However, the results of figure 4 would indicate that even though the most potent glycan found on the surface of red cells is the type II H antigen, the YSD library did contain antibody clones that were barely detectable in the lamprey plasma.

-A point of concern in terms of usability of these reagents is found in **lines 343-355.** More specifically, it is clear that the “type of linker that couples the sugar to the slide” and other “presentation” details significantly impacts the binding of a given reagent. It may turn out that **the glycan ligand for one of these reagents is not recognized within a biological sample (due to presentation differences),** leading

to the potential for false negatives. While this should not impede publication, suggestions for mitigating this should be included in the Discussion.

***Response of Authors:** In our opinion, this point would apply to both monoclonal antibodies and lectins commonly used in glycobiology. It is important to remember that glycan microarrays are a screening platform to initially determine specificity, but due to the heterogeneity of the glycan structures present in natural tissues, antigen recognition must always be orthogonally validated using alternative methods to determine the true epitope. We have addressed this comment on line 476-487 in the discussion.*

-Minor: The statement on line 221-222 about self-tolerance and processing should have the **appropriate citations**.

***Response of Authors:** We thank the reviewer and have added the following citations for this statement: (17,40) on line 218.*

-Minor: The authors discuss the advantages of the lamprey system over mIgM on lines 248-252. Isn't one of the other advantages **that the lamprey molecules are more specific (less cross-reactive)?**

***Response of Authors:** Yes, the lamprey monoclonal VLRBs that have been characterized are more specific than many of the murine IgM antibodies. However, we were cautious on making too broad of a statement here, as a direct comparison between monoclonal VLRBs and mIgMs was not made.*

REVIEWERS' COMMENTS:

Reviewer #1 (Remarks to the Author):

The authors addressed all the comments and the manuscript can be accepted for publication in its current format.

Reviewer #2 (Remarks to the Author):

After reading the author's responses and their revised manuscript, I feel that the authors sufficiently addressed the concerns of the reviewers. I remain enthusiastic about this body of work being of high interest to the community.

Reviewer #3 (Remarks to the Author):

The authors have addressed my concerns well enough to support publication of this important work. There is certainly more to be done in this system to understand all of the key characteristics, but it would be inappropriate to ask for more in a single article.

Response to Revision Review- McKittrick et al

In order to accept your paper, we require the following:

* The attached document detailing your point-by-point response to editorial requests and any issues raised by our referees. This document should be uploaded as a 'Related Manuscript File' entitled "response to editorial requests" or similar. An additional cover letter is optional.

Response: *This file contains our point-by-point responses to reviewers and editorial requests.*

* The final version of your text as a Word or TeX/LaTeX file, with any tables prepared using the Table menu in Word or the table environment in TeX/LaTeX and using the 'track changes' feature in Word. Please note that we cannot accept BibTeX files. References should be included within the manuscript file itself. Authors who wish to use BibTeX to prepare their references should therefore copy the reference list from the .bbl file that BibTeX generates and paste it into the main manuscript .tex file (and delete the associated \bibliography and \bibliographystyle commands).

Response: *We have uploaded the revised manuscript as a Word file with track changes and with references included.*

* Production-quality versions of all figures, supplied as separate files. Figures divided into parts should be labelled with a lowercase bold a, b, and so on. To ensure the swift processing of your paper please provide the highest quality, vector format, versions of your images (.ai, .eps, .psd) where available. Text and labelling should be in a separate layer to enable editing during the production process. If vector files are not available then please supply the figures in whichever format they were compiled (do not save as flat .jpeg or .TIFF files). Any chemical structures or schemes contained within figures should additionally be supplied as separate ChemDraw (.cdx) files. If your artwork contains any photographic images, please ensure these are at least 300 dpi.

Response: *We have uploaded the individual figures for the manuscript.*

* The final version of any Supplementary Information (figures, tables, notes etc) in one PDF file. See the attached document for details.

** Please note that Supplementary Information cannot be changed after the paper has been accepted **

Response: *We have uploaded the final version of the Supplementary materials.*

*Reproducibility checklists: See the attached document for details.

Response: *We have included the relevant checklists.*

* If you wish, an interesting image (but not an illustration or schematic) for consideration as a 'Featured Image' on the Communications Biology homepage. The file should be 1400x400 pixels in RGB format and should be uploaded as 'Related Manuscript File'. In addition to our home page, we may also use this image (with credit) in other journal-specific promotional material. If you submit a suggested featured image, please also include a completed image License to Publish form.

Congratulations on an excellent paper!

Response: *We thank the Editor for their positive comments and help in handling our manuscript.*

REVIEWERS' COMMENTS:

Reviewer #1 (Remarks to the Author):

The authors addressed all the comments and the manuscript can be accepted for publication in its current format.

Response: *We thank the reviewer for their positive endorsement of our manuscript.*

Reviewer #2 (Remarks to the Author):

After reading the author's responses and their revised manuscript, I feel that the authors sufficiently addressed the concerns of the reviewers. I remain enthusiastic about this body of work being of high interest to the community.

Response: *We thank the reviewer for their positive endorsement of our manuscript and enthusiasm for our work.*

Reviewer #3 (Remarks to the Author):

The authors have addressed my concerns well enough to support publication of this important work. There is certainly more to be done in this system to understand all of the key characteristics, but it would be inappropriate to ask for more in a single article.

Response: *We thank the reviewer for their positive endorsement and support of our manuscript.*

Response to Second Editorial Comments, McKittrick et al

In checking your recent submission to Communications Biology, we have found the following problems, which we must ask you to address before we can formally consider your manuscript.

1) Please provide us with more information on the creation of Figure 3 and Figure 7. Was this image and every element of this image created by you and/or your co-Authors? Was it created using any previously-created elements? If the image is from a database please supply proof of permission for its use (receipt, express permission from creator, confirmation of compatible Open Access license or Public Domain).

Response: *All of the images in Figure 3 and Figure 7 were taken by, or drawn by the authors of the manuscript. Figure 7 did contain stock images (top right panel), that were meant to be placeholders. This was an oversight and they have been removed. Our sincere apologies for this mistake. A detailed description of who created each component of the image is given below:*

Figure 3. mFc blue image drawn by RDC; VLRB cartoon is O-13 apo structure taken from pdb file (5UE1) and redrawn to change color scheme in pymol by TRM. Yeast cartoon originally drawn by BRH and modified slightly by TRM, yeast images on the array taken at Emory University in RDC lab by TRM. Glycan cartoons drawn by TRM.

Figure 7.

Top left panel:

Glycan microarray cartoon, ELISA plate, red blood cells with sugar, blue and green antibodies drawn by RDC. VLRB schematic drawn by TRM using O-13 apo structure

Bottom left panel:

Glycan microarray, red blood cells drawn by RDC. Bacterial plate drawn by TRM, FACs plot is from the biotinylated red blood cell lysate enrichment of the type O library. This data is not included in the manuscript, but can be provided if necessary.

Top right panel:

CHO cells with glycans drawn by TRM, RBCs drawn by RDC, both the pig lung and human milk images were removed.

Bottom middle panel:

DNA cartoons drawn by TRM, yeast cartoon originally drawn by BRH and modified slightly by TRM, antibody and fluorophore drawn by RDC.

Bottom right panel:

DNA, flask drawn by TRM, VLRB cartoon is O-13 apo structure taken from pdb file (5UE1) and redrawn to change color scheme in pymol. Fc cartoon drawn by RDC.

2) Please cite all panels in your figure legends. Currently, figure 1 does not cite all panels (a-g), and Supplementary figure 1 does not cite panels (a-b).

Response: *This has been completed.*

3) Please provide your Author Cover Letter to the editor, and your Rebuttal Letter to Reviewers, as two separate items. Rebuttal letter should only contain reviewer comments and responses.

Response: *This has been completed and 2 files uploaded.*

4) Please provide descriptions for each supplementary data file (1-3) in your author cover letter or as separate files.

Response: *These legends are provided in a separate file and are given below:*

Supplementary file 1 contains all of the lamprey plasma and mouse serum run on the CFG array. Each individual tab is labeled by the antibody type and immunogen. Note: $a/b = \alpha/\beta$ linkage.

Supplementary file 2 contains all of the data collected from the yeast bound to the array. This includes the libraries as well as the RBC36 clone. Each tab is labeled by the sample. Note: $a/b = \alpha/\beta$ linkage.

Supplementary file 3 contains all of the data from each monoclonal antibody on the CFG array and NCFG Sialyl Derivative array. Each tab is labeled by sample and array type. Note: $a/b = \alpha/\beta$ linkage.

5) Due to our policy on Transparent Peer Review, you must clearly state in your cover letter whether you wish to "OPT IN" or "OPT OUT" to the publication of the reviewer reports.

Response: *This has been completed.*

6) Please remove supplementary figure legends from your article file. The legends included in your supplementary materials file is sufficient.

Response: *This has been completed and the legends are at the beginning of the supplementary materials file (legends and figures).*

7) Please ensure the Supplement sentence in supplementary figure 1 fits on the page. Currently, it goes off the end of the PDF file.

Response: *This has been completed.*

8) Please ensure your supplementary figure citations match between the article file and figure legends. Currently supplementary figures 4 and 5 are cited as "supplementary figure S4 and S5".

Response: *This has been completed.*